# Ephrin-B1 regulates the adult diastolic function through a late postnatal maturation of cardiomyocyte surface crests

Clement Karsenty[1,2], Celine Guilbeau-Frugier[1,3], Gaël Genet[4], Marie-Helene Seguelas[1], Philippe Alzieu[5], Olivier Cazorla[6], Alexandra Montagner[1], Yuna Blum[7], Caroline Dubroca[8], Julile Maupoint[8], Blandine Tramunt[1,9], Marie Cauquil[1], Thierry Sulpice[8], Sylvain Richard[6], Silvia Arcucci[1], Remy Flores-Flores[1], Nicolas Pataluch[1], Romain Montoriol[3], Pierre Sicard[6], Antoine Deney[1], Thierry Couffinhal[5,10], Jean-Michel Senard[1,11], Celine Galés[1]*

[1]INSERM, UMR 1297, Institut des Maladies Métaboliques et Cardiovasculaires, Université de Toulouse, Toulouse, France; [2]Department of Pediatric Cardiology, Centre Hospitalier Universitaire de Toulouse, Toulouse, France; [3]Department of Forensic Medicine, Centre Hospitalier Universitaire de Toulouse, Université de Toulouse, Toulouse, France; [4]Department of Cell Biology, University of Virginia School of Medicine, Charlottesville, United States; [5]Université de Bordeaux, INSERM, Biologie des maladies cardiovasculaires, Pessac, France; [6]Université de Montpellier, INSERM, CNRS, PhyMedExp, Montpellier, France; [7]IGDR UMR 6290, CNRS, Université de Rennes 1, Rennes, France; [8]CARDIOMEDEX, Escalquens, France; [9]Department of Diabetology, Metabolic Diseases & Nutrition, Centre Hospitalier Universitaire de Toulouse, Toulouse, France; [10]Service des Maladies Cardiaques et Vasculaires, CHU de Bordeaux, Bordeaux, France; [11]Department of Clinical Pharmacology, Centre Hospitalier Universitaire de Toulouse, Toulouse, France

*For correspondence:
celine.gales@inserm.fr

**Abstract** The rod-shaped adult cardiomyocyte (CM) harbors a unique architecture of its lateral surface with periodic crests, relying on the presence of subsarcolemmal mitochondria (SSM) with unknown role. Here, we investigated the development and functional role of CM crests during the postnatal period. We found in rodents that CM crest maturation occurs late between postnatal day 20 (P20) and P60 through both SSM biogenesis, swelling and crest-crest lateral interactions between adjacent CM, promoting tissue compaction. At the functional level, we showed that the P20-P60 period is dedicated to the improvement of relaxation. Interestingly, crest maturation specifically contributes to an atypical CM hypertrophy of its short axis, without myofibril addition, but relying on CM lateral stretching. Mechanistically, using constitutive and conditional CM-specific knock-out mice, we identified ephrin-B1, a lateral membrane stabilizer, as a molecular determinant of P20-P60 crest maturation, governing both the CM lateral stretch and the diastolic function, thus highly suggesting a link between crest maturity and diastole. Remarkably, while young adult CM-specific Efnb1 KO mice essentially exhibit an impairment of the ventricular diastole with preserved ejection fraction and exercise intolerance, they progressively switch toward systolic heart failure with 100% KO mice dying after 13 months, indicative of a critical role of CM-ephrin-B1 in the adult heart function. This study highlights the molecular determinants and the biological implication of a new late P20-P60 postnatal developmental stage of the heart in rodents during which, in part, ephrin-B1 specifically regulates the maturation of the CM surface crests and of the diastolic function.

## Editor's evaluation

This article will be of interest to the field of cardiac development and function. It provides detailed characterization of the cardiomyocyte surface crests, and also provides evidence that loss of Ephrin-B1 leads to compromised surface crest formation and diastolic dysfunction. The article did not provide direct evidence that the diastolic dysfunction of Ephrin-B1 cardiac mutant is primarily attributable to the compromised postnatal maturation of cardiomyocyte surface crests; however, this work and the rigor applied represent an important hypothesis and contribution to the field.

## Introduction

The mammalian adult cardiomyocyte (CM) harbors a typical rod shape specifically dedicated to the function of the adult heart. Understanding the maturation of the adult rod-shaped CM can provide important insights for regenerative medicine, especially for the differentiation of hiPSC cells into fully mature rod-shaped adult CMs, which still remains an obstacle. To date, the molecular events leading to the setting of the CM rod shape during the postnatal stage are still unknown.

Indeed, despite fetal and adult CMs being extensively studied, the postnatal stage has only recently garnered an unprecedented interest with the discovery of the potential of adult CMs to proliferate and regenerate the heart (*Bergmann et al., 2009*). Thus, strategies in this field have focused on boosting this regenerative process by exploiting factors specifically involved in the early CM proliferation arrest that occurs during the postnatal period (*Karra and Poss, 2017*). However, despite the postnatal maturation period characterizing the exit of the CM from the cell cycle, it also coincides with a morphogenesis step during which the CM switches from a proliferative rounded shape to a nonproliferative mature rod shape (*Anversa et al., 1980*). In situ, mammalian CMs stop dividing and start their morphogenesis maturation around postnatal day 7 (P7) (*Porrello et al., 2011*; *Porrello et al., 2013*).

The postnatal morphogenesis stage of the CM relies on a polarization process that results in the asymmetric organization of the plasma membrane components underlying specific functions with an atypical basolateral polarity. It begins with a longitudinal elongation process of the CM to progressively evolve into the rod-shaped characteristic of the mature adult state. Contrary to epithelial cells harboring a basolateral and apical side, the adult CM lacks the apical side and looks like a barrel surrounded by a unique basal side with a basement membrane connecting the fibrillar extracellular matrix (ECM) and flanked by two intercalated disks (ID) involved in the CM-CM tight interactions, shaping the longitudinal alignment of myofibers within the tissue (*Kanzaki et al., 2010*; *Kanzaki et al., 2012*). The ID phenocopies the lateral side of polarized epithelial cells with the presence of tight, adherent, gap junctions and desmosomes that play a specific role in the anchorage of the contractile myofilaments but also in the synchronization of the contraction. One architectural feature of the basal side of the adult CM, also called the lateral membrane, also relies on its important intracellular invaginations into transverse T-tubules, which play a key role in both action potential propagation and $Ca^{2+}$ handling/excitation–contraction coupling *Ibrahim et al., 2011*. Likewise, the extracellular side of the lateral membrane seems more complex than initially suspected since, besides the presence of a costamere structure expressing classical transmembrane receptors (integrin and DGC dystroglycan-glycoprotein complex) that connect the ECM components to the intracellular myofibrils, we and others have reported the atypical presence of transmembrane proteins, claudin-5 and ephrin-B1 (*Genet et al., 2012*; *Sanford et al., 2005*), more likely involved in cell–cell communication. This finding was unexpected since, contrary to the ID, the lateral membrane was until recently viewed as a CM side lacking physical interactions with neighboring CMs. However, these last years, using high-resolution nanoscale imaging, i.e., SCIM, AFM, and MET, we and Gorelik's lab have described a highly organized architecture of the lateral membrane on adult CMs with periodic crests (*Dague et al., 2014*; *Guilbeau-Frugier et al., 2019*; *Nikolaev et al., 2010*) filled with subsarcolemmal mitochondria (SSM) whose role is unknown. Furthermore, we provided evidence for the existence, in the 3D cardiac tissue, of intermittent lateral crest–crest contacts all along the lateral membrane through claudin-5/claudin-5 tight junctional interactions (*Guilbeau-Frugier et al., 2019*), thus reconciling the presence of such cell–cell communication proteins on the lateral face of the adult CM. However, exactly when and how the lateral membrane crests of the CM maturate is completely unknown.

In this study, we investigated the maturation of the CM crests and their role during the postnatal period. We provide evidence for a late postnatal development stage, following the set-up of the rod shape, during which crests fully mature through SSM swelling. We also show that this maturation step of the crests is ephrin-B1-dependent and specifically regulates the diastolic function of the adult heart.

## Results

### CM lateral surface crests maturate late after postnatal day 20

In-depth study of the CM crests is only meaningful when studied in situ, i.e., within the cardiac tissue, since (1) we have previously shown that these structures rapidly shrink and disappear following CM isolation and culture (*Dague et al., 2014*; *Guilbeau-Frugier et al., 2019*) and (2) we have demonstrated that CMs establish intermittent lateral interactions through physical crest–crest interactions (*Guilbeau-Frugier et al., 2019*), an observation unique to the cardiac tissue since adult CMs loss their physical contacts in culture. Also, we investigated crest maturation during the postnatal period on the left ventricle tissue of male rat hearts from different postnatal days (P0/birth, P5, P20, P60/young adult) using transmission electron microscopy (TEM) as previously described (*Guilbeau-Frugier et al., 2019*). We first focus our attention on the rat species for facilitating visualization of the CM crests/ SSM at early postnatal stages. Surprisingly, at P0, while the contractile apparatus is disorganized in the neonatal CM, myofibrils are already orientated on the longitudinal axis of the cell with the first layer already anchored to the plasma membrane through Z-lines, and thus outlining a periodic crest-like architecture, more likely plasma membrane protrusions, which can yet be visualized all along the CM surface (*Figure 1A*, *Figure 1—figure supplement 1*). These immature and disordered crests already attempt to interact with crests from a neighboring CM. At this stage, no SSM could be observed. Then, crest maturation throughout the postnatal period occurs in two steps. A first early step, already completed by P5, relies on the maturation of the Z-lines, which delimits each sarcomere from the outer myofibril, allowing better visualization of the surface crest structure, concomitant to the myofilament alignment/organization that follows the morphological elongation of the CM (*Figure 1A*, P5). However, crests still display an unstructured morphology and their heights, directly correlated with the SSM number as previously described in adult CMs (*Guilbeau-Frugier et al., 2019*), are small ('flat'-appearance) due to the lack or the presence of very small SSM, likely immature SSM (*Figure 1A*). By comparison, much larger but disorganized interfibrillar mitochondria (IFM) can be visualized in CMs at birth (P0) predominantly around CM nuclei (*Figure 1—figure supplement 1*), while they mature until P60 through both swelling (completed at P20) and alignment along the myofilaments (*Figure 1—figure supplement 2*). The crest immaturity persists until P20, a late stage of the postnatal period, while the CM has already implemented its rod shape (*Figure 1—figure supplement 3*) and completed its whole cytoarchitecture (*Piquereau et al., 2010*), as indicated by the perfect alignment of sarcomere Z-lines between myofibril layers inferred from the α-actinin staining (*Figure 1—figure supplement 4*). It is worth noting that at P20, CMs from left ventricular myocardium display different morphologies with both rod-shaped and spindle-shaped CMs when compared with only rod-shaped CMs at P60 (*Figure 1—figure supplement 3*, lower panels). Interestingly, a second but delayed maturation step of the surface crests occurs between P20 and the adult stage (P60) through substantial SSM swelling (*Figure 1A*), during which crest heights significantly expand up, correlating with an increase in both the SSM number and area (*Figure 1B*). Similar P20-P60 crest/SSM maturation was depicted in the mouse heart tissue (*Figure 1—figure supplement 5*), thus agreeing with conservation of the postnatal maturation of the heart in mouse and rat as recently reported (*Li et al., 2022*). We further confirmed this maturation of the CM surface crests after P20 by analyzing the expression and localization of claudin-5, a tight-junctional protein that we previously described as a determinant of the lateral crest–crest interactions between the lateral membrane of neighboring CMs (*Guilbeau-Frugier et al., 2019*). While claudin-5 protein expression is progressively induced in the cardiac tissue at birth (P0), reaching its maximal expression at P5-P10 (*Figure 1C*, left panel), complete localization of the protein at the lateral membrane of the CM occurs only between P20 and P60 (*Figure 1C*, right panel), most likely in agreement with the implementation of claudin-5/claudin-5-dependent interactions necessary to clip crests from neighboring CMs that we previously described at the adult stage (*Guilbeau-Frugier et al., 2019*). In line with this mechanism, atypical tight junctions connecting crests

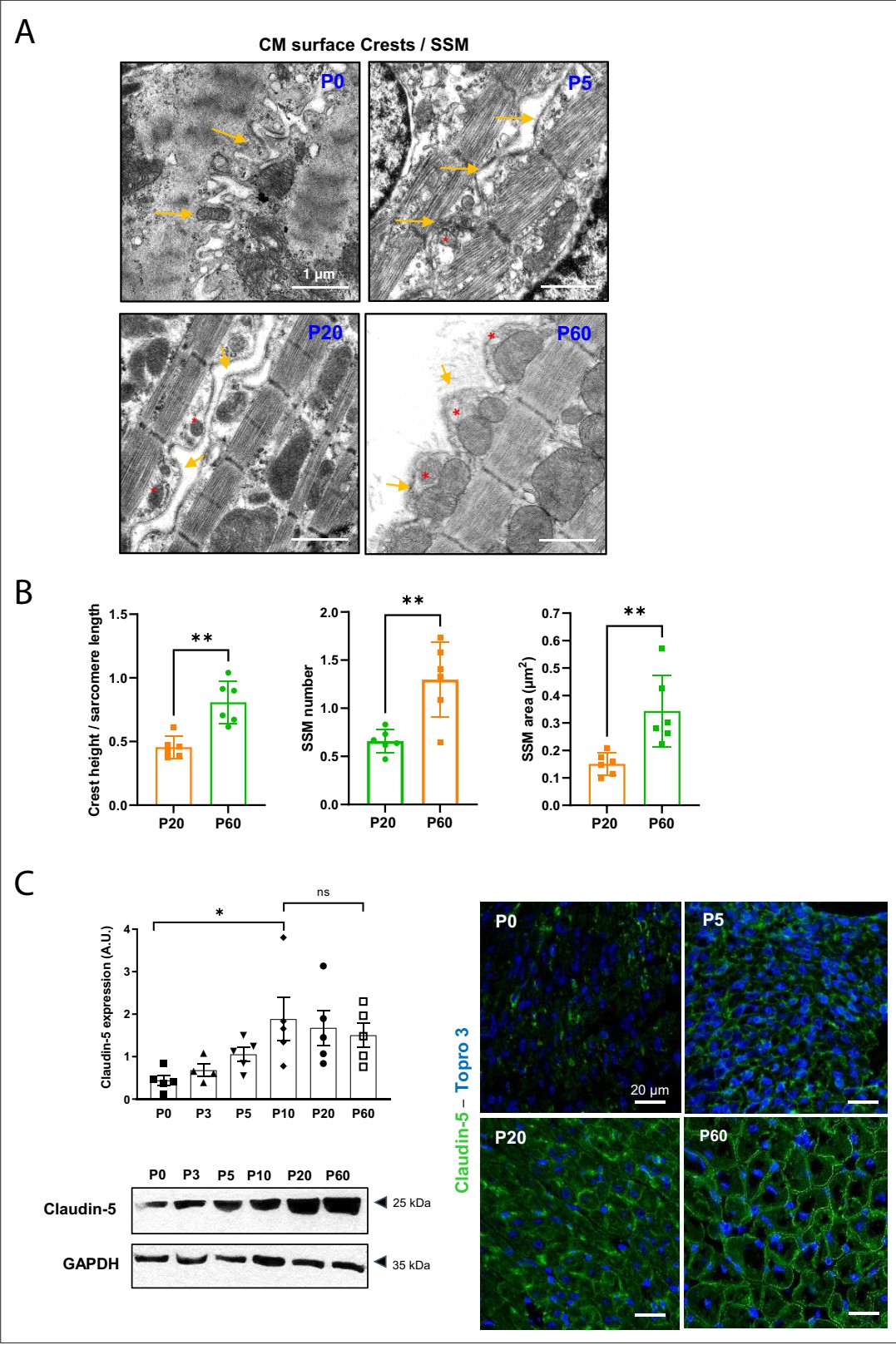

**Figure 1.** Cardiomyocyte (CM) surface crests mature after postnatal day 20 (P20). (**A**) Transmission electron microscopy (TEM) micrographs showing representative CM surface crest relief (yellow arrows) and associated subsarcolemmal mitochondria (SSM) (red stars) during rat postnatal maturation at postnatal day 0 (P0), 5 (P5), 20 (P20), and 60 (P60). (**B**) Quantification of crest heights/sarcomere length (left panel), SSM number/crest (middle

*Figure 1 continued on next page*

*Figure 1 continued*

panel), and SSM area (right panel) from TEM micrographs obtained from P20- or P60 rat hearts (P20 or P60 n = 6 rats; 4–8 CMs/rat, ~70 crests/rat). (**C**) (Left panel) Western blot quantification of claudin-5 protein expression in heart tissue from P0 to P60 old rats (upper panel) and representative immunoblot (lower panel) (P0 n = 5, P3 n = 4; P5 n = 5, P10 n = 5, P20 n = 5, P60 n = 5); (right panel) immunofluorescent localization of claudin-5 in heart cryo-sections from P0, P5, P20, and P60 rats. Data are presented as mean ± SD. Unpaired Student's *t*-test for two group comparisons and one-way ANOVA for claudin-5 Western -blot analysis, Tukey post-hoc test for six group comparisons with P0 as control. *p<0.05, **p<0.01. ns, not significant.

The online version of this article includes the following source data and figure supplement(s) for figure 1:

**Source data 1.** Raw data of CM crest maturity quantified in P20 and P60 rats.

**Source data 2.** Raw data of claudin-5 western blot (quantifications).

**Source data 3.** Raw data of claudin-5 western-blot (Original films).

**Figure supplement 1.** Ultrastructure of the neonatal cardiomyocyte (CM) at birth.

**Figure supplement 2.** Interfibrillar mitochondria (IFM) during postnatal maturation.

**Figure supplement 3.** Maturation of cardiomyocyte (CM) morphology during the postnatal period.

**Figure supplement 4.** Organization of cardiomyocyte (CM) myofibrils during the postnatal period.

**Figure supplement 5.** Cardiomyocyte (CM) surface crests mature after postnatal day 20 (P20) in mice.

**Figure supplement 5—source data 1.** Raw data of CM crest maturation between P20 and P60 in mice quantified on TEM images.

**Figure supplement 6.** Neighboring cardiomyocytes (CMs) establish direct lateral physical contacts through crest–crest interactions only at the adult stage.

**Figure supplement 7.** Maturation of the intercalated disk of cardiomyocytes (CMs) during the postnatal period.

**Figure supplement 8.** Organization of the T-tubule network during the late postnatal period.

**Figure supplement 8—source data 1.** Raw data of TT power and TT frequency.

from the lateral face of neighbor CMs could be observed only in the P60 adult stage (*Figure 1—figure supplement 6*).

Remarkably, the maturation of the CM lateral surface occurring between P20 and P60, at least in part through the setting of crest–crest interactions, occurs concomitantly with an ultimate maturation of both the ID and the T-tubules, relying on a spatial reorganization of their specific components. Thus, while connexin 43 (gap junctions), desmoplakin 1/2 (desmosomes), and N-cadherin (Adherens junctions) are located on both the lateral membrane and the ID at P20, they fully relocalize to the ID at P60 (*Figure 1—figure supplement 7*). Likewise, RyR and caveolin-3, T-tubule markers (see 'Methods'), are still misaligned at P20 while a perfect alignment along the sarcomere Z-lines on the short CM axis can be observed at P60 (*Figure 1—figure supplement 8A and B*). In agreement, the TT power (TT regularity) tends to increase after P20 while the TT periodicity (frequency) is already established at P20 (*Figure 1—figure supplement 8C* and 'Methods').

Overall, these results indicate that the architecture of the CM plasma membrane as a whole still continues to mature late after P20, following the establishment of the rod shape.

## Evidence for a late P20-P60 postnatal developmental stage of the mammalian heart dedicated to the development of the diastolic function

The existence of a late postnatal maturation stage between P20 and P60 of the mammalian heart was further confirmed in mice by transcriptional analysis of left ventricular tissue. Volcano plot analysis of the transcriptome and the heat map clearly show a significant difference between P20 and P60 heart gene expression (*Figure 2A and B*) with 1000 protein-coding genes that are up- or down-regulated between P20 and P60 (p<0.05, fold change > 1.5). The gene ontology (GO) enrichment analysis revealed significantly affected biological pathways (p<0.05) (*Figure 2A*, *Figure 2—figure supplement 1*) that are upregulated and mainly related to processes of the immune defense system, muscle cell differentiation, angiogenesis, positive regulation of cell death, different metabolisms including antioxidant defense, plasma membrane-related transport/signaling together with many pathways related to the nervous system development. By contrast, many ECM and developmental

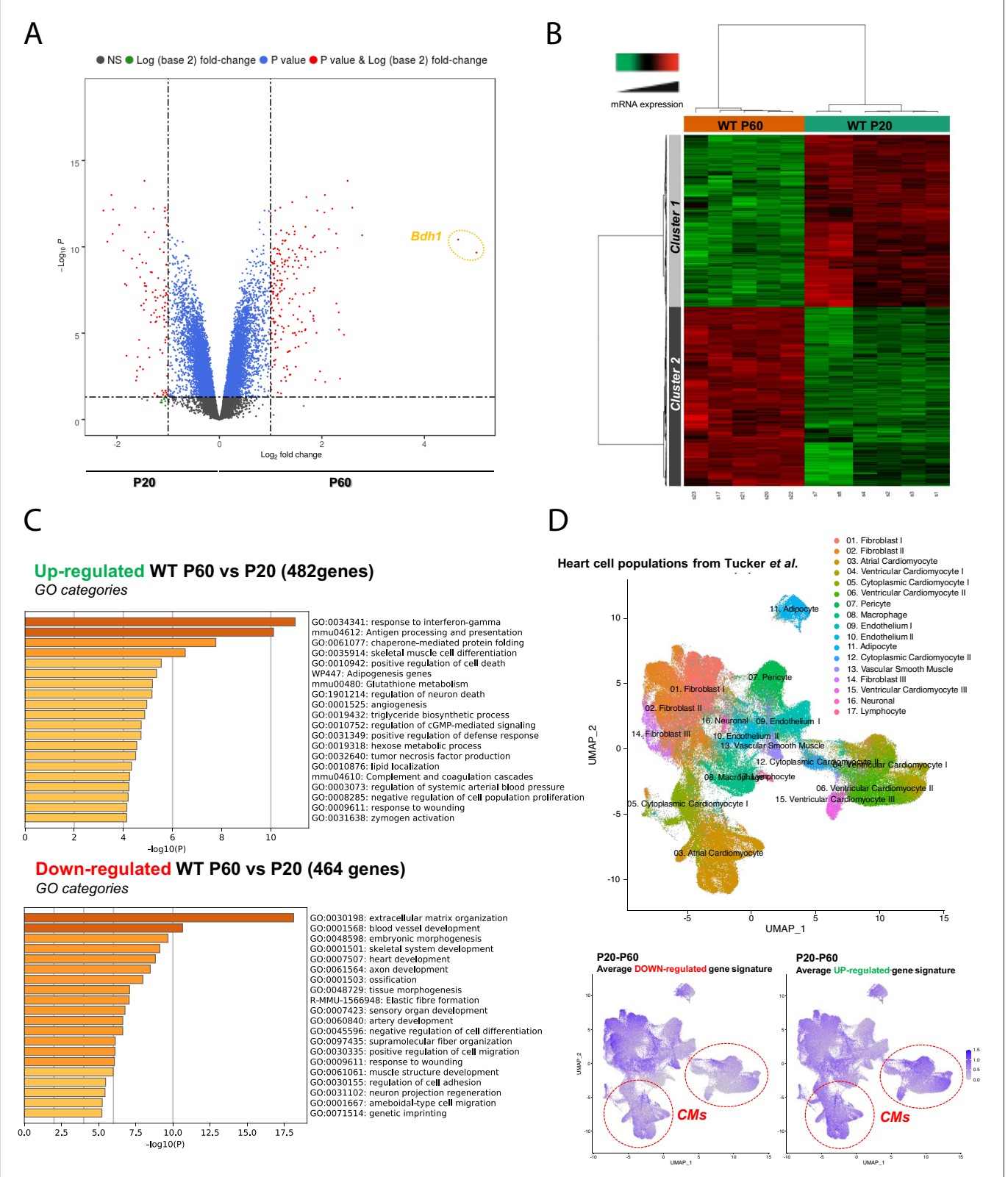

**Figure 2.** Late postnatal maturation stage between P20 and P60 of the mammalian heart confirmed by transcriptional analysis of left ventricular tissue. (**A**) Volcano plot of differences in gene expression between P20 and P60 mice. Colors indicate p<0.05 and log (base 2) fold change >2 (red), p<0.05 and log (base 2) fold change <2 (blue) and nonsignificant (NS) (black). (**B**) Heatmap presenting data from a microarray experiment performed with heart samples (P20 n = 6, P60 n = 5). Hierarchical clustering is also shown, which allows the definition of two gene clusters (p≤0.05). (**C**) Gene ontology

*Figure 2 continued on next page*

*Figure 2 continued*

(GO) analysis of upregulated (upper panel) or downregulated (lower panel) genes between P60 and P20. The false discovery rate is provided for each category. (**D**) Uniform manifold approximation and projection (UMAP) plot displaying cellular diversity present in the human heart using Tucker et al.'s single-cell RNA-seq dataset. Each dot represents an individual cell. (Upper panel) Colors correspond to the cell identity provided by the authors. The average expression of downregulated- (left lower panel) or upregulated (right lower panel) gene signatures of left ventricles between P20 and P60 rats was calculated for each cell population and represented on the UMAP plot. Color key from gray to blue indicates relative expression level from low to high.

The online version of this article includes the following figure supplement(s) for figure 2:

**Figure supplement 1.** Signaling pathways upregulated or downregulated during the late postnatal period in left ventricles from WT mice.

**Figure supplement 2.** Regulation of cardiomyocyte (CM) and metabolic markers during the late postnatal period.

processes are downregulated. It is worth noting that some upregulated metabolic pathways relate to heart fuels other than fatty acids and glucose (tryptophan, glutamate, glycine/serine/threonine metabolisms), thus suggesting that the heart increases its metabolic flexibility between P20 and P60. To get a better idea of the cardiac cells impacted by the P20-P60 transition, we next performed a gene clustering according to the whole cardiac cell populations established by *Tucker et al., 2020* in the human heart ('Materials and methods'). This analysis has to be taken only from a global qualitative standpoint since we assumed between murine and human heart (1) similar populations of the largest cardiac cells (cardiomyocytes, endothelial cells, fibroblasts, etc.), (2) similar main transcriptomic features of these largest cardiac cells, and (3) similar cardiac cell populations at P20 and P60. In line with the GO analysis, we found that the main downregulation of the transcriptome between P20 and P60 occurs primarily in the fibroblast populations known to be involved in the synthesis of the fibrillar ECM but also to a lesser extent in the CM populations (*Figure 2D*), consistent with the skeletal system and heart development pathways. By contrast, although all cardiac cells seem to be involved in the transcriptional maturation, the P20-P60 upregulated transcriptome largely referred to the ventricular CM populations compared to the downregulated one (*Figure 2D*), in agreement with the evidence of a prominent muscle cell differentiation pathway depicted through the GO analysis. The existence of an important transcriptional maturation step of the CM during the P20-P60 postnatal window is further reinforced by the modulation of key CM-specific protein-encoding genes, especially several proteins from the contractile apparatus or regulating the contractile machinery at the lateral membrane (*Figure 2—figure supplement 2A*). Another CM maturation also occurs at the metabolic level while the metabolic switch of the heart from glycolysis to fatty acid (FA) oxidation was already established during the early postnatal period (*Lopaschuk and Jaswal, 2010*), with the remarkable upregulation of the *BDH1* gene that encodes the β-hydroxybutyrate dehydrogenase, the limiting mitochondrial enzyme for ketone body (KB) uptake during fatty acid catabolism, but also to a lesser extent the upregulation of key actors in the glycolytic metabolism (*PFKFB2, SLC2A4*) (*Figure 2—figure supplement 2B*). Overall, these results indicate that not only the CM but the whole heart still continue developing after P20.

To better understand the functional impact of the late P20-P60 maturation of the heart and for technical suitability, we evaluated both the systolic and diastolic functions of rat hearts during this postnatal stage. Longitudinal echocardiographic evaluation reveals a similar left ventricular ejection fraction (LVEF) in both P20 and P60 rats (*Figure 3A*), indicating earlier maturation of the systolic function during the postnatal period. By contrast, we observed specific changes in the diastolic function between P20 and P60 as measured by noninvasive Doppler imaging and showing a significant increase in passive filling (E/A) and an improvement in relaxation (*decrease in the isovolumic relaxation time [IVRT], increase in the e'/a' and the early diastolic mitral annular tissue velocity e', without change in the LV filling pressures E/e'*) (*Figure 3A*), most likely indicating that the diastolic function of the rat heart maturates during the late postnatal period, between P20 and P60. Further confirming the P20-P60 set-up of the adult diastolic function, while P20 and P60 rats display similar heart rates, invasive left ventricle catheter analysis shows increased systolic and diastolic blood pressure (SBP/DBP) as well as end-diastolic pressure (EDP) and an improvement in diastolic relaxation reflected by the dP/dt$_{min}$ and the decreased time constant of isovolumic relaxation (Tau) (*Figure 3B*). These data indicate that the late P20-P60 developmental stage of the heart is specifically required for the maturation of the diastolic function.

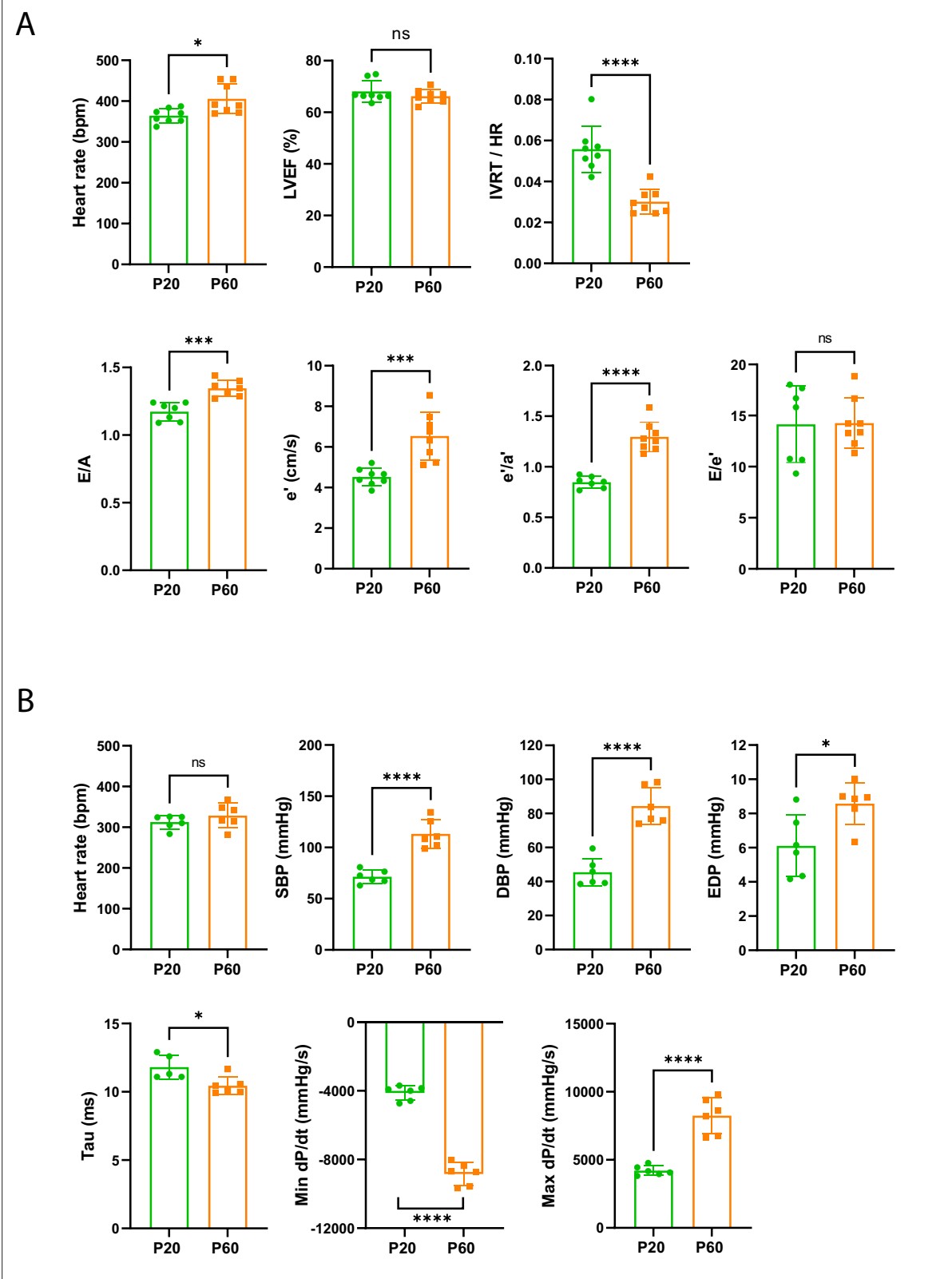

**Figure 3.** Late cardiac development is associated with the maturation of diastolic function. (**A**) Echocardiography performed on P20 and P60 rats using M-mode, B-mode, Doppler flow, and tissular Doppler to measure systolic and diastolic function. Left ventricular ejection fraction (LVEF), isovolumetric relaxation time (IVRT), E/A ratio, e' peak velocity, e'/a' ratio, and E/e' ratio were measured (P20 or P60 n = 8 rats). (**B**) Cardiac catheterization to assess the systolic and diastolic function of P20- or P60 rats. Systolic and diastolic blood pressure (SDB and DBP), end diastolic LV pressure (EDP), Tau and min

*Figure 3 continued on next page*

*Figure 3 continued*

and max dP/dt were measured (P20 or P60 n = 6 rats). Data are presented as mean ± SD. Unpaired Student's *t*-test for two group comparisons *p<0.05, **p<0.01, ***p<0.001, ****p<0.0001 ns, not significant.

The online version of this article includes the following source data for figure 3:

**Source data 1.** Raw data of echocardiography/doppler measurements in P20 or P60 rats.

**Source data 2.** Raw data of cardiac catheterization in P20 and P60 rats.

## New atypical P20-P60 physiological CM hypertrophy through lateral stretching

To better understand the P20-P60 maturation of the crests/SSM at the lateral face of the CMs, we next focused our attention on the more global maturation of the CM during this specific P20-P60 transition. We observed that CMs from the left ventricle of rat hearts undergo significant hypertrophy between P20 and P60 as indicated by the substantial increase in their cross-sectional area (*Figure 4A*), which peaks at P45 (*Figure 4—figure supplement 1*), and in both their long and short axes (*Figure 4B*, *Figure 4—figure supplement 2*). Similar late hypertrophy of CMs was observed in mice (*Figure 4—figure supplement 3*). This P20-P60 CM hypertrophy was confirmed by echocardiography with a significant increase in the left ventricle posterior wall thickening (LVPWd) and cavity size (LVEDV, LVEDD) (*Figure 4C*), all indicative of an overall heart growth. Surprisingly, this physiological CM hypertrophy is atypical since it is not correlated with an expected increase in the myofilament compartment as indicated by the constant number of CM myofibrils between P20 and P60 (*Figure 4D*). This is corroborated by a marked decrease in the heart weight to body weight ratio during this period (*Figure 4E*), as previously reported (*Piquereau et al., 2010*). Interestingly, we also noticed a significant and specific increase in the sarcomere heights with no variation of the sarcomere lengths (*Figure 4F*) which was inversely correlated with a decrease in the inter-lateral space between two CMs (*Figure 4G*), likely indicative of a lateral stretch of the CMs and a cardiac tissue compaction. Further supporting the CM lateral stretch that should distend the myofibrils, we observed larger distances between the thick myosin filaments at P60 than at P20 (*Figure 4—figure supplement 4*).

Taken together, these results highlight a new type of physiological cardiac hypertrophy that occurs during late postnatal development and that presumably relies, at least in part, from the lateral vantage point, on the stretching of the CM lateral membrane.

## *Efnb1*-specific knockdown in the CM impairs the late maturation of surface crests and the diastolic function

We have previously shown within the tissue that the CM hypertrophy through lateral stretching between P20 and P60 occurs concomitantly with the maturation of the crests that coat the whole CM lateral surface and, more specifically, with the implementation of the crest–crest lateral interactions between neighboring CMs. This P20-P60 late maturation stage also correlates with the implementation of the adult diastolic function. Interestingly, we have previously hypothesized that the CM surface crests, a hallmark of the adult stage in mammals, might play a specific role in the control of the diastole but not the systole, based on a hypothetic model in which crest height and lateral crest–crest interaction limit the sarcomere maximal extension during relaxation (*Guilbeau-Frugier et al., 2019*). Altogether, these observations raise questions about a specific contribution of the crest maturation, including the setting of crest–crest interactions with intermittent tight junctions, in the lateral stretch of the CM that could fine-tune the adult diastole. Hence, we next asked whether the maturation of the CM surface crests occurring after P20 could directly control the CM lateral hypertrophy and the diastolic function of the heart.

For that purpose, we examined the role of ephrin-B1, a transmembrane protein that we previously identified as a new protein of the lateral membrane of the adult CM independent from the integrin or the dystroglycan systems (*Genet et al., 2012*). More interestingly for the P20-P60 maturation, we demonstrated that ephrin-B1 stabilizes the adult rod-shape of the CM through specific regulation of its lateral membrane overall structure through an hypothetic mechanical stretching, the mechanism of which was unsolved. However, we showed that ephrin-B1 directly interacts with claudin-5, another atypical protein at the lateral membrane of the adult CM, and controls its expression (*Genet et al., 2012*). Interestingly, we showed that young adult mice harboring a CM-specific deletion of *Efnb1* did

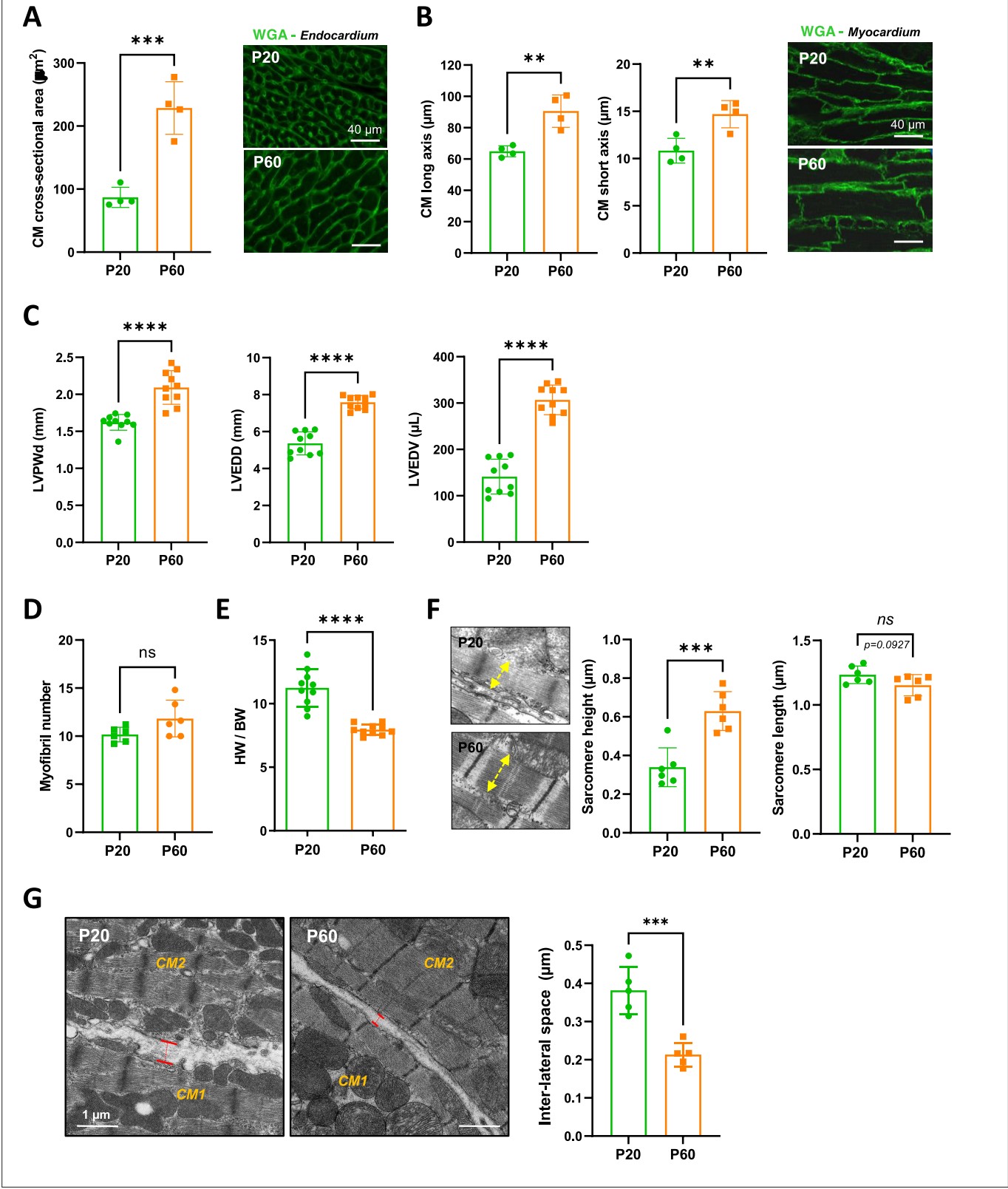

**Figure 4.** Evidence for a late postnatal maturation stage of the cardiomyocyte (CM) and the heart. (**A**) (Left panel) CM area quantification from wheat-germ agglutinin (WGA)-stained heart cross-sections (endocardium) obtained from P20 and P60 rats (~400 CMs/rat; n = 4 rats per group), as illustrated in the right panel (**B**) (Left panel) CM long- and short-axis quantification from WGA-stained heart cross-sections (myocardium) obtained from P20 and P60 rats (~200 CMs/rat; n = 4 rats per group) and illustrated in the right panel. (**C**) Analysis of echocardiography-based morphometry of hearts from

*Figure 4 continued on next page*

*Figure 4 continued*

P20 and P60 rats (P20 or P60 n = 10 rats). (**D**) Myofibril number quantified on the longitudinal CM axis from TEM micrographs of cardiac tissue from P20 or P60 rats (P20 or P60 n = 6 rats; 4–8 CMs/rat). (**E**) Heart weight/body weight ratio of P20 or P60 rats (P20 or P60 n = 10 rats). (**F**) (Left panel) TEM micrographs showing representative sarcomere stretch (left arrows) from P20 to P60 rat; (right panel) quantification of sarcomere height (P20 or P60 n = 6 rats; 4–8 CMs/rat, ~35 sarcomeres/rat). (**G**) (Left panel) TEM micrographs showing representative lateral membrane space between two neighboring CMs (red arrows) in cardiac tissue from P20 or P60 rats; (right panel) quantification of the lateral membrane interspace (P20 or P60 n = 6 rats; 4–8 CMs/rat, ~35 lateral interspaces/rat). Data are presented as mean ± SD. Unpaired Student's *t*-test for two group comparisons *p<0.05, **p<0.01, ***p<0.001, ****p<0.0001, ns, not significant.

The online version of this article includes the following source data and figure supplement(s) for figure 4:

**Source data 1.** Raw data of CM area quantification.

**Source data 2.** Raw data of CM long- and short- axis quantifications.

**Source data 3.** Raw data of echocardiography-based morphometry of hearts.

**Source data 4.** Raw data of CM myofribrils quantification in TEM images.

**Source data 5.** Raw data of Heart weight/body weight ratios.

**Source data 6.** Raw data of sarcomere height and length quantification.

**Source data 7.** Raw data of lateral interspace quantifications.

**Figure supplement 1.** Kinetics of cardiomyocyte (CM) hypertrophy during the late postnatal period.

**Figure supplement 1—source data 1.** Raw data of CM area quantifications.

**Figure supplement 2.** Late postnatal growth of isolated cardiomyocytes (CMs).

**Figure supplement 2—source data 1.** Raw data of CM long an short axis quantifications.

**Figure supplement 3.** Hypertrophy of cardiomyocytes (CMs) in mice during the late postnatal period.

**Figure supplement 3—source data 1.** Raw data of CM area quantifications.

**Figure supplement 3—source data 2.** Raw data of CM short and long axis quantifications.

**Figure supplement 4.** Late postnatal maturation of the myofibrils.

not demonstrate contractile defects, but the diastolic function was not studied. By contrast, these animals demonstrated a high susceptibility to cardiac stresses. Also, given the importance of claudin-5 in the setting of the crest–crest lateral interactions within the adult cardiac tissue (*Guilbeau-Frugier et al., 2019*) and its regulation by ephrin-B1, we next explore the potential role of ephrin-B1 as a specific determinant of CM surface crests and the diastole.

We first studied ephrin-B1 expression/localization in the cardiac tissue from rat hearts during the early postnatal development. As shown in *Figure 5A* and similarly to what we observed for claudin-5, ephrin-B1 reaches maximal expression very early during the postnatal period (P5). However, the complete trafficking of ephrin-B1 from the cytosol to the CM surface is only achieved at P60. To further explore the role of ephrin-B1 during the late postnatal developmental stage, we took advantage of a knock-out mouse model harboring a CM-specific deletion of *Efnb1* (KO) that we previously described but only at the young adult stage (*Genet et al., 2012*). Indeed, while 2-month-old young adult CM-specific *Efnb1* KO mice were highly sensitive to cardiac stresses, we did not find any major cardiac dysfunction at resting state (*Genet et al., 2012*). It is worth noting that ephrin-B1 does not contribute to major cardiac developmental processes before P20 since P20 CM-specific *Efnb1* KO mice and WT mice demonstrate almost similar left ventricular transcriptome as highlighted by the volcano plot (only six genes different between the two genotypes) (*Figure 5—figure supplement 1*). At the cellular level, the lack of ephrin-B1 in the CM partially impairs the P20-P60 physiological hypertrophy of the CM (*Figure 5B*), without modification of the myofibril number (*Figure 5C*). Consistent with ephrin-B1-specific expression at the lateral membrane, *Efnb1* deletion more specifically impedes the lateral stretch (short axis) of the CM but not the longitudinal one (*Figure 5D*) and accordingly leads to a decrease in both the sarcomere heights (*Figure 5F*) and the tissue compaction (increased lateral interspace) (*Figure 5G*). Of note, *Efnb1* deletion only partially prevents CM hypertrophy (partial CM short-axis elongation), most likely due to the other molecular events taking place at the lateral membrane during the P20-P60 stage, i.e., the interactions with the ECM (integrin, dystroglycan complex) that we previously showed to be independent of ephrin-B1 (*Genet et al., 2012*). At the lateral membrane level, P20 CM-specific *Efnb1* KO and WT mice harbor similar unstructured

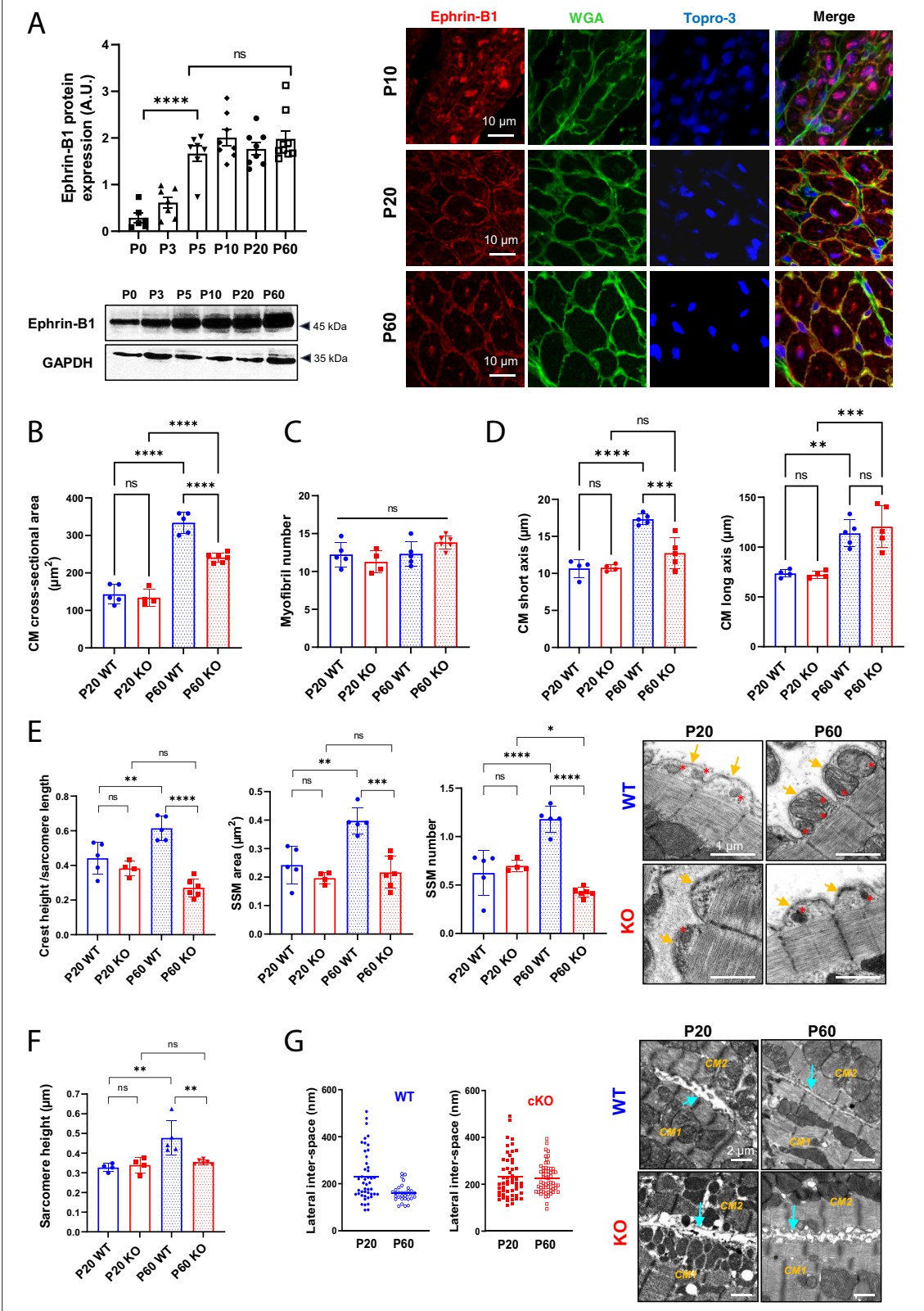

**Figure 5.** *Efnb1*-specific knockdown in the cardiomyocyte (CM) impairs the late maturation of CM surface crests. (**A**) (Left panel) Western blot quantification of ephrin-B1 protein expression in heart tissue from P0 to P60 rats (upper) and representative immunoblot (lower) (P0 n = 6, P3 n = 7; P5 n = 7, P10 n = 8, P20 n = 8, P60 n = 8 rats); (right panel) immunofluorescent localization of ephrin-B1 (white arrows) in heart cryosection from P10, P20, and P60 old rats. At P10, ephrin-B1 (red) was mainly expressed in CM nuclei. At P20, ephrin-B1 is expressed at the CM lateral membrane but still highly in the

*Figure 5 continued on next page*

*Figure 5 continued*

CM cytoplasm while at P60, the protein is mainly expressed in the lateral membrane. Nuclei were stained using topro-3 (blue). (**B**) CM area quantification from WGA-stained heart cross-sections (endocardium) obtained from P20 or P60 *Efnb1*[CM-spe]KO and WT mice (~120 CMs/mice; WT P20 n = 5, cKO P20 n = 4, WT P60 n = 5, cKO n = 6 mice). (**C**) Myofibril number quantification from TEM micrographs of cardiac tissue from P20 or P60-old *Efnb1*[CM-spe]KO and WT mice (WT P20 n = 5, cKO P20 n = 4, WT P60 n = 5, cKO n = 6 mice; 4–8 CMs/mouse). (**D**) CM long and short axis quantified from WGA-stained heart cross-sections (myocardium) obtained from P20 or P60 *Efnb1*[CM-spe]KO and WT mice (WT P20 n = 4, cKO P20 n = 4, WT P60 n = 5, cKO n = 5 mice; ~30 CMs/mouse). (**E**) Quantification of crest heights/sarcomere length (left panel), subsarcolemmal mitochondria (SSM) area (middle panel) and SSM number/crest (right panel) from TEM micrographs obtained from P20- or P60 *Efnb1*[CM-spe]KO and WT mice (WT P20 n = 5, cKO P20 n = 4, WT P60 n = 5, cKO n = 5 mice, ~60 crests per group) and illustrated in the right panel (arrow = surface crest; red star = SSM). (**F**) Quantification of sarcomere heights from TEM micrographs obtained from P20 or P60 *Efnb1*[CM-/-] KO and WT mouse hearts (P20 WT n = 4, P20 cKO n = 4, P60 WT n = 5; P60 cKO n = 6 mice; 4–8 CMs/mouse, ~50 sarcomeres/mouse). (**G**) (Left panels) Quantification of the lateral membrane interspace between two neighboring CMs from TEM micrographs obtained from P20 or P60 *Efnb1*[CM-/-] KO and WT mouse hearts (P20 WT n = 1, P20 cKO n = 2, P60 WT n = 3; P60 cKO n = 3 mice; ~40 lateral membrane spaces/mouse) and (right panel) representative TEM. One-way ANOVA, Tukey post-hoc test for six group comparisons with P0 as control. ****p<0.0001 for ephrin-B1 Western blot analysis. Data are presented as mean ± SD. Unpaired Student's *t*-test for two group comparisons or two-way ANOVA with Tukey post-hoc test for four group comparisons *p<0.05, **p<0.01, ***p<0.001, ****p<0.0001. ns, not significant.

The online version of this article includes the following source data and figure supplement(s) for figure 5:

**Source data 1.** Raw data of ephrin-B1 quantifcation in western-blot.

**Source data 2.** Raw data of CM area quantifcations.

**Source data 3.** Raw data of myofibril number quantications.

**Source data 4.** Raw data of CM long and short axis quantifications.

**Source data 5.** Raw data of CM crest maturity quantified in P20 and P60 WT or KO mice.

**Source data 6.** Raw data of sarcomere heights quantified in P20 and P60 WT or KO mice.

**Source data 7.** Raw data of lateral interspace quantified in P20 and P60 WT or KO mice.

**Source data 8.** Original film from western-blot analysis of ephrin-B1 expression.

**Figure supplement 1.** Left ventricle transcriptome differences between P20 *Efnb1*[CM-/-]KO and WT mice.

surface crests with immature SSM but only WT mice underwent crest maturation at P60 (*Figure 5E*), thus demonstrating a key role for ephrin-B1 in the postnatal maturation of the surface crest/SSM.

Given the specific impact of *Efnb1* deletion on the P20-P60 CM lateral stretch and on the crest/SSM maturation, we next assessed the role of ephrin-B1 in the cardiac function of young adult male mice (P60/2 months) with a specific focus on the diastolic function that we never explored before (*Genet et al., 2012*). As previously reported (*Genet et al., 2012*), we did not notice differences in heart rate and LVEF between 2-month-old WT and CM-specific *Efnb1* KO mice (*Figure 6A*). This result is more likely consistent with a preserved function of the IFM subpopulation in the CM-specific *Efnb1* KO mice, spatially more specifically dedicated to the myofibril contractility (*Hollander et al., 2014*). However, and consistent with the role of ephrin-B1 in the lateral stretch-based hypertrophy of the CM, CM-specific *Efnb1* KO mice display a significant decrease in LVPWd conversely to an increase in left ventricular internal diameter (left ventricular internal diameter end diastole [LVIDd]) and volume (LV end-diastolic volume [LVEDV]) (*Figure 6A*), in agreement with our previous study (*Genet et al., 2012*). More interestingly, compared with WT mice, CM-specific *Efnb1* KO mice display significantly elongated IVRT, increased left atrial volume (left atrial to aortic root ratio [LA/Ao]), and decreased E/A as well as heterogeneous E/E' (*Figure 6A*), all indicative of impaired LV relaxation. Despite preservation of the LVEF, we also measured a significant decrease in ventricular global longitudinal strain (LV-GLS) in KO mice (*Figure 6A*) reflecting abnormal longitudinal systolic function. Diastolic dysfunction in KO mice was further confirmed by cardiac catheterization since 2-month-old CM-specific *Efnb1* KO mice exhibit increased EDP and heterogeneous dP/dt but with similar relaxation time constants (tau) compared with age-matched WT mice (*Figure 6B*). The diastolic defects of CM-specific *Efnb1* KO mice rely on a specific impairment of the CM relaxation since both the diastolic basal sarcomere length (SL) shortening and relaxation velocities are significantly reduced in isolated intact CMs from CM-specific *Efnb1* KO compared to WT mice (*Figure 6C*). Furthermore, consistent with heart failure with preserved ejection fraction (HFpEF) phenotype in which CM contraction defects coexist with the relaxation impairment (*Methawasin et al., 2016*; *Primessnig et al., 2016*; *Schiattarella et al., 2019*), the auxotonic contraction indexed by the SL shortening as well as the SL shortening velocities are also significantly decreased in CMs from CM-specific *Efnb1* KO compared to WT CMs (*Figure 6C*).

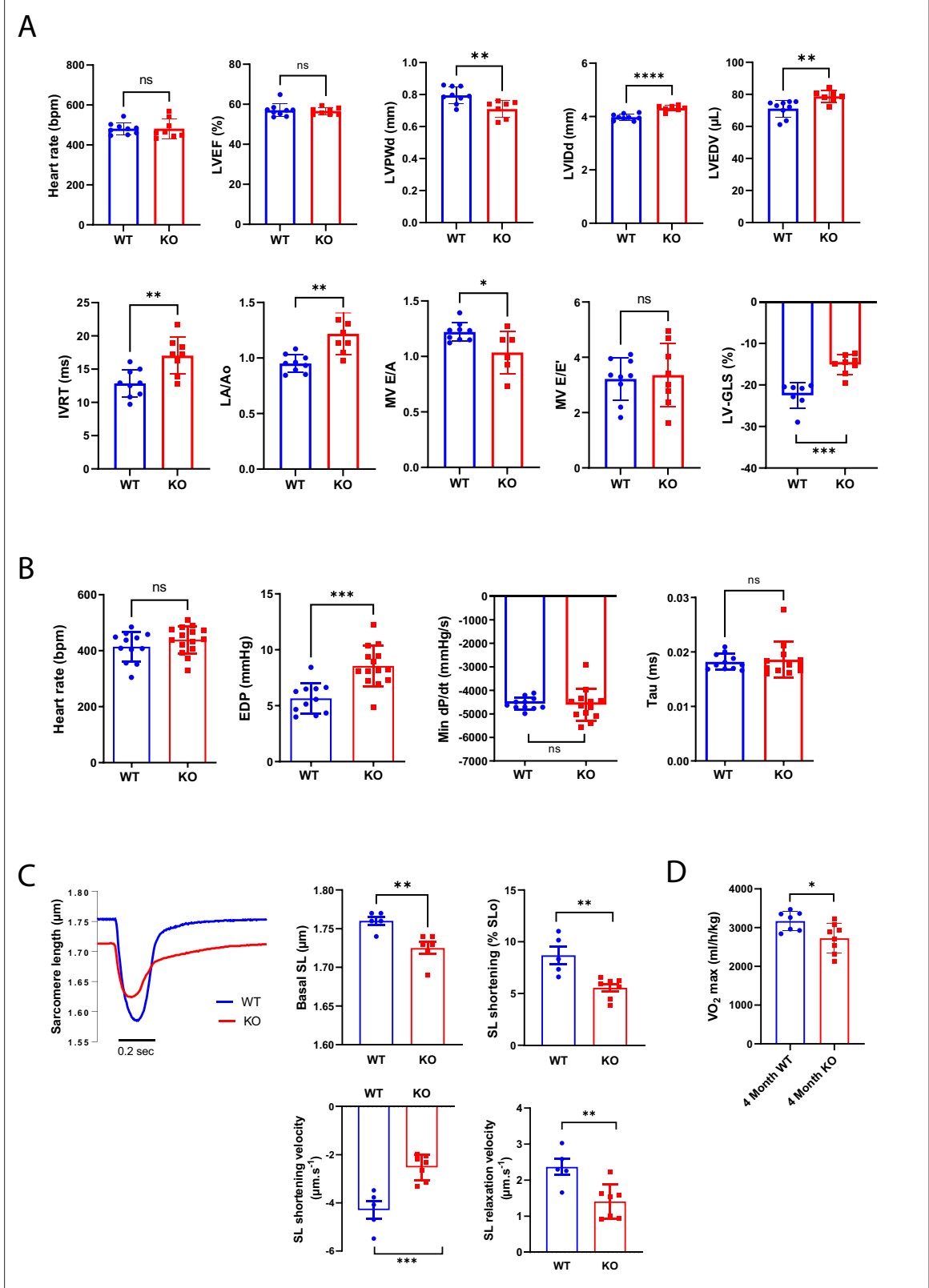

**Figure 6.** Constitutive *Efnb1*-specific knockdown in the cardiomyocyte (CM) impairs diastolic function. (**A**) Echocardiography performed on P60 *Efnb1*^CM-/-^KO or WT mice with 2D, TM, Doppler flow, and tissular Doppler analysis to measure morphometry, systolic and diastolic function. Left ventricular ejection fraction (LVEF), interventricular septum wall thickness in diastole (IVSd), left ventricular internal diameter end diastole (LVIDd), left ventricular end-diastolic volume (LVEDV), isovolumetric relaxation time (IVRT), left atrium/aorta ratio (LA/Ao), E/A ratio, e' peak velocity, e'/a' ratio, and E/e' ratio

*Figure 6 continued on next page*

*Figure 6 continued*

were measured (WT n = 9, cKO n = 8 mice). (**B**) Cardiac catheterization to assess diastolic function of P60 *Efnb1$^{CM-/-}$*KO or WT mice, measuring end diastolic LV pressure (EDP), Tau, and min dP/dt (WT n = 12, cKO n = 14 mice). (**C**) (Lleft panel) Representative contraction evoked by electrical field stimulation as measured from sarcomere length (SL) shortening in isolated CMs (left ventricles) from P60 *Efnb1$^{CM-spe}$*KO or WT mice; (middle panel) basal sarcomere length (SL); (right panel) sarcomere length (SL) shortening during contraction; (lower-left panel) sarcomere length shortening velocity; (lower-right panel) sarcomere length relaxation velocity (WT n = 5, cKO n = 7 mice). (**D**) Treadmill exercise tolerance assay assessed by the VO$_{2max}$ measured from 4-month-old *Efnb1$^{CM-spe}$*KO or WT mice (WT n = 7, cKO n = 8 mice). Data are presented as mean ± SD. Unpaired Student's *t*-test for two group comparisons *p<0.05, **p<0.01, ***p<0.001. ****p<0.0001. ns, not significant.

The online version of this article includes the following source data for figure 6:

**Source data 1.** Raw data of echocardiography/doppler analysis in P60 WT or KO mice.

**Source data 2.** Raw data of measurements from LV cardiac catherization in P60 WT and KO mice.

**Source data 3.** Raw data of contraction measurements after electrical stimulation of isolated CMs.

**Source data 4.** Raw data of VO2max measurements in treadmill exercise tolerance assay.

Finally, CM-specific *Efnb1* KO mice also display significant exercise intolerance compared with WT mice, as indicated by the significant decrease in maximum oxygen uptake (*Figure 6D*). Together, these results indicate that young adult male CM-specific *Efnb1* KO mice recapitulate some features of clinical HFpEF, thus demonstrating a pivotal role for ephrin-B1 in the control of the diastolic function. In line with this assumption, a recent RNAseq study performed on ventricles from HFpEF, heart failure with reduced ejection fraction (HFrEF), and control patients reported a specific and more significant downregulation of the *Efnb1* gene in HFpEF than HFrEF patients (p=9.10$^{-14}$ HFpEF vs. control, p=2.10$^{-5}$ HFrEF vs. control, p=0.005 HFpEF versus HFrEF) (*Hahn et al., 2021*).

Despite constitutive CM-specific *Efnb1* KO and WT mice displaying quite similar transcriptomic features at P20, we cannot completely conclude that the diastolic phenotype of the KO mice at the adult stage directly arises from a defect of the CM crest maturation and not indirectly from another earlier event during the cardiac development. To discriminate between these two possibilities, we have now used a tamoxifen-inducible conditional-knock-out (αMHC-Mer-Cre-Mer) of *Efnb1* in the CM. In agreement with most conditional mouse models, injection of tamoxifen for four consecutive days in 2-month-old *Efnb1*$^{flox/flox}$-αMHC-Cre$^+$ (cKO) and their *Efnb1*$^{flox/flox}$-αMHC-Cre$^-$ littermates (Ctrl) led to a specific but partial deletion (~50 %) of ephrin-B1 in CMs from the cKO mice only when analyzed after 1 month following the last tamoxifen treatment (*Figure 7—figure supplement 1*). In similar experimental conditions (*Figure 7A*), we next examined whether this CM-specific deletion of *Efnb1* at the adult stage impacts on the CM crest architecture and the diastolic function. As shown in *Figure 7B*, deletion of *Efnb1* at the adult stage led to partial loss of the CM surface crests in cKO mice compared to their control littermates. It should be noticed that, in these experiments, control mice do not harbor 100% CMs with crests (*Figure 7B*, left panel), most likely reflecting a tamoxifen influence. Likewise *Efnb1* KO mice, cKO mice display a significant left ventricle dilatation as reflected by the increase in the left ventricular internal diameter (LVIDd) associated with a trend to the increase in the left ventricle volume (LVEDV), a lack of left ventricle hypertrophy (IVSd, LVPWd), and a preserved ejection fraction (LVEF) (*Figure 7C*). More interestingly, the deletion of *Efnb1* at the adulthood still promotes significant diastolic defects since cKO mice show significant elongated IVRT with no modification of the E/A together with a significant increase in EDP, heterogeneous dP/dt but similar relaxation time constants compared with their control littermates (*Figure 7D*).

Altogether, these results clearly demonstrate that ephrin-B1 controls the maturation of the CM surface crests during the P20-P60 postnatal period and contributes to the regulation of the adult diastolic function.

## *Efnb1*$^{CM-/-}$ mice switched progressively from HFpEF to HFrEF and all died at 14 months of age due to T-tubule disorganization

Finally, we monitored changes in the cardiac function of CM-specific *Efnb1* KO and WT mice in medium and long term. Interestingly, while in young adulthood (2 months) CM-specific *Efnb1* KO mice display LV diastolic dysfunction with preserved ejection fraction compared to WT mice, LVEF decreased progressively over time in these mice, which start developing moderate HFrEF at 9 months, evolving toward severe HFrEF by 13 months (*Figure 8A*) and 100% mortality after 15 months (*Figure 8B*).

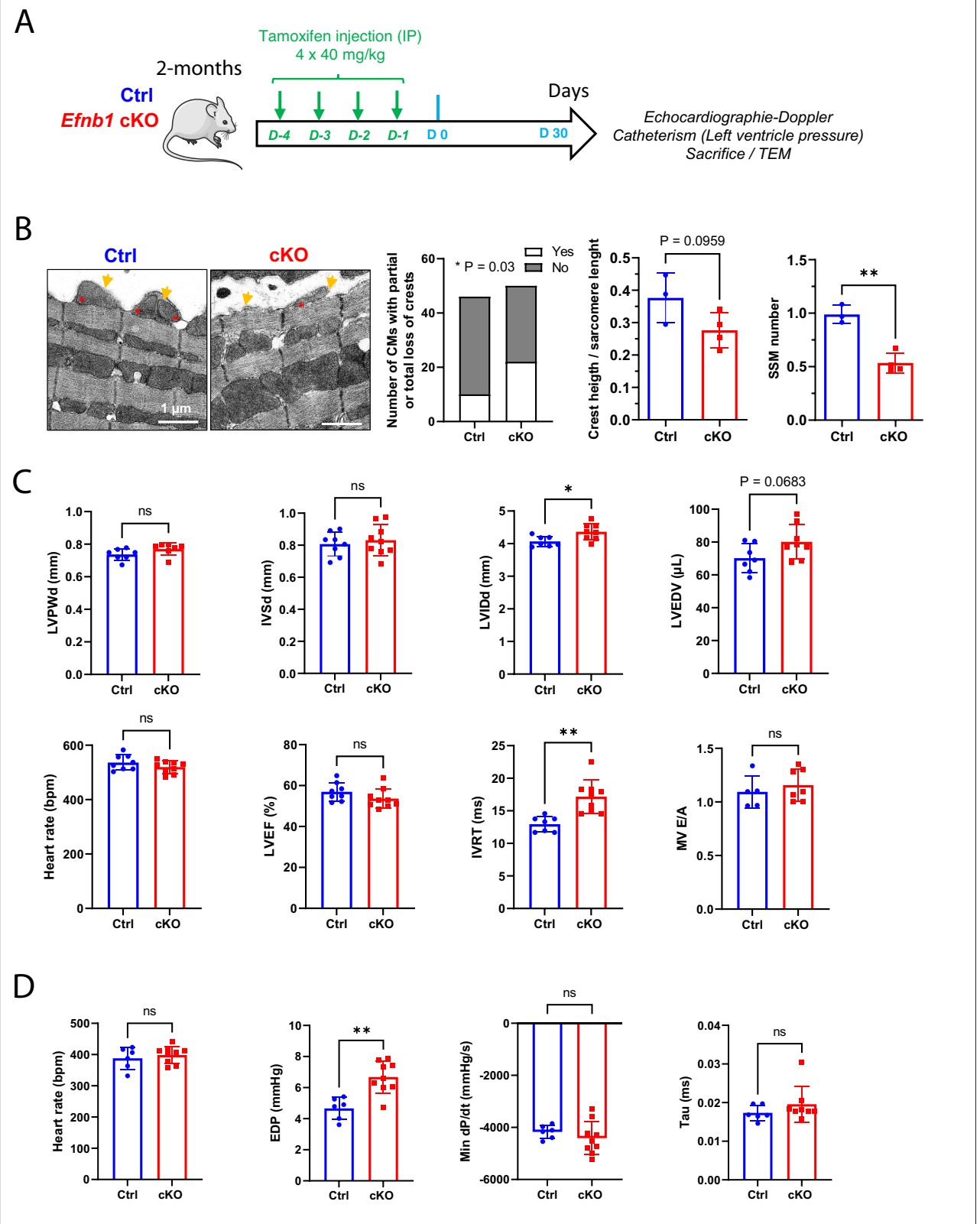

**Figure 7.** Conditional *Efnb1*-specific knockdown in the cardiomyocyte (CM) at the adult stage impairs CM crest maturation and the diastolic function. (**A**) Two-month-old Ctrl (*Efnb1*flox/flox-αMHC-Cre⁻) and *Efnb1* cKO (*Efnb1*flox/flox-αMHC-Cre⁺) mice were treated with tamoxifen for four consecutive days and subjected 1 month later to echocardiography, measurements of left ventricle pressures and phenotyping of CM surface crests from left ventricles. (**B**) Qualitative evaluation of CM number exhibiting total or partial subsarcolemmal mitochondria (SSM loss) (left panel), quantification of crest heights/

*Figure 7 continued on next page*

*Figure 7 continued*

sarcomere length (middle panel), and SSM number/crest (right panel) from TEM micrographs obtained from Ctrl or *Efnb1* cKO (Ctrl n = 3, cKO n = 4 mice; ~60 crests per group) and illustrated in the left panel (arrow = surface crest; asterisk = SSM). (**C**) Echocardiography performed on Ctrl or *Efnb1*cKO mice with 2D, TM, Doppler flow, and tissular Doppler analysis to measure morphometry, systolic and diastolic function. Left ventricular ejection fraction (LVEF), left ventricular posterior wall thickness in end diastole (LVPWd), interventricular septum wall thickness in end diastole (IVSd), left ventricular internal diameter end diastole (LVIDd), left ventricular end-diastolic volume (LVEDV), and isovolumetric relaxation time (IVRT) (WT n = 8, cKO n = 9 mice). (**D**) Cardiac catheterization to assess diastolic function of Ctrl or *Efnb1* cKO mice, measuring end diastolic LV pressure (EDP), Tau, and min dP/dt (Ctrl n = 6, cKO n = 9 mice). Fisher's exact test for (**B**, left panel). Data are presented as mean ± SD. Unpaired Student's *t*-test for two group comparisons *p<0.05, **p<0.01. ns, not significant.

The online version of this article includes the following source data and figure supplement(s) for figure 7:

**Source data 1.** Raw data of CM crest maturity quantified in control and cKO mice.

**Source data 2.** Raw data of echocardiography/doppler analysis in adult control or cKO mice.

**Source data 3.** Raw data of measurements from LV cardiac catherization in control and cKO mice.

**Figure supplement 1.** Characterization of αMHC-MerCreMer[+/-]-*Efnb1*[flox/flox] mice (cardiomyocyte [CM]-specific inducible conditional *Efnb1* KO, *Efnb1* cKO).

**Figure supplement 1—source data 1.** Raw data of ephrin-B1 expression quantified in western-blot.

Accordingly, LVIDd and LVEDV significantly increase over time only in KO mice (*Figure 8A*) concurrently with the development of compensatory cardiac hypertrophy (IVSd, *Figure 8A, C and D*) and fibrosis around 12 months of age (*Figure 8E*). Corroborating the HF progression in CM-specific *Efnb1* KO mice, while T-tubule misalignment from Z-lines (HFrEF marker) is already detectable but limited to discrete local regions in the cardiac tissue of some KO CMs at 2 months of age, this disorganization progressively spreads over time to all CMs after 12 months compared with WT mice (*Figure 8F*).

Collectively, these data indicate that the lack of ephrin-B1 in CM primarily impairs the diastolic function of young adult mice, which progressively switch toward a systolic defect with aging. Hence, specific defects in the maturation of the CM surface crests during the late postnatal development that will early impact the diastolic function of the adults, more likely asymptomatic in young adulthood, will be only revealed later in life through the onset of the systolic dysfunction.

## Discussion

In this study, we describe a new developmental stage occurring in the late postnatal period between P20 and P60, during which crests of the CM lateral membrane maturate through SSM biogenesis, swelling, and crest–crest interactions within the tissue through an ephrin-B1/claudin-5 mechanism, thus allowing CM lateral stretching and a global compaction of the cardiac tissue (*Figure 9*). We demonstrate that this mechanism regulates the adult diastolic function. Hence, these results shed light on the molecular mechanism by which ephrin-B1 regulates the CM lateral membrane that we described before (*Genet et al., 2012*). Taken together, our findings identify crest subdomains of the adult CM lateral surface as novel specific determinants of cardiac diastole.

Postnatal maturation of the heart in mammals has been paid much less attention than embryonic and fetal development. This would be a prerequisite for future pediatric clinical trials, which are seriously lacking. It also contributes to a significant knowledge gap that impedes research progression in regenerative medicine since this postnatal maturation coincides with the proliferation blockage of the CM. To date, at the cellular level, postnatal cardiac maturation has been essentially described until P20 in rodents, most likely related to the implementation of the typical adult rod shape of the CM. Very few studies have examined the specific P20-adult stage window, assuming that the genetic maturation of the heart is completed by P20 and that the heart merely undergoes a substantial growth beyond P20 until the young adult stage (*Piquereau et al., 2010*). A recent study analyzed early gene expression of rodent hearts and of CMs during different postnatal days, including P21 and P56 (young adult), and while the authors clearly identified different transcriptomic signatures between these two periods, they focused their study on the P1-P7 transition (*Li et al., 2022*). However, this P20-Adult period coincides with the weaning time in rodents (~P21) and thus with a critical nutritional/metabolic switch in the organism likely to influence heart function. So far, research has focused on the P0 (birth) to P20 postnatal window, during which the heart switches from a hyperplasic growth (CM proliferation) (~P3-P5) to an hypertrophic growth (increase in CM size) (P20-21) (*Bishop et al., 2021*; *Leu*

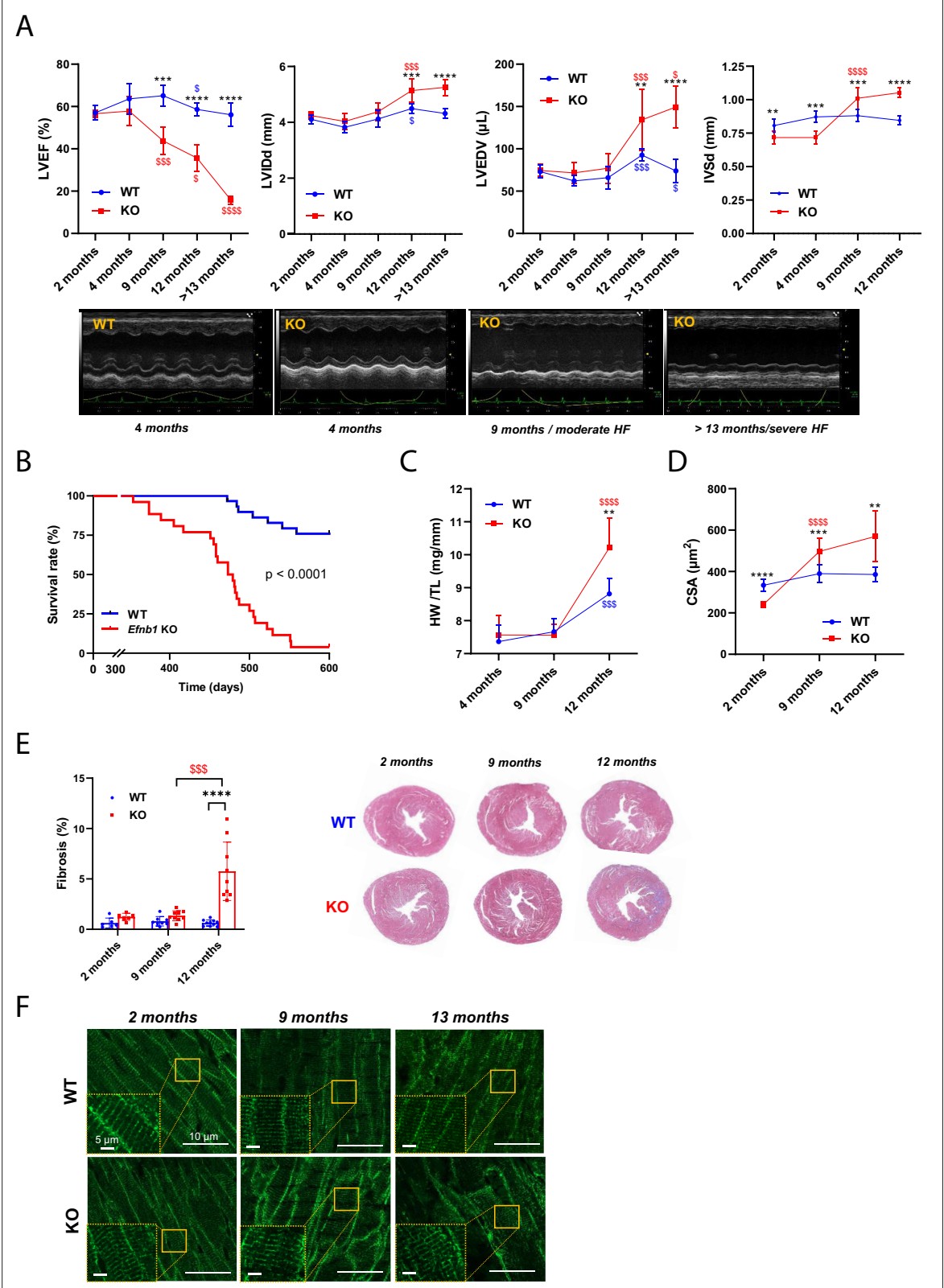

**Figure 8.** *Efnb1*^CM-/- KO mice switch progressively from heart failure with preserved ejection fraction (HFpEF) to heart failure with reduced ejection fraction (HFrEF). (**A**) (Upper panels) Longitudinal echocardiography to assess morphometry and systolic function of *Efnb1*^CM-/-KO and WT mice over time. Left ventricular ejection fraction (LVEF), left ventricular internal diameter end diastole (LVIDd), left ventricular end diastolic volume (LVEDV), and interventricular septum wall thickness in diastole (IVSd) were measured in 2-, 4-, 9-, 12- and >13-month-old mice (WT 2 months n = 9, 4 months

*Figure 8 continued on next page*

*Figure 8 continued*

n = 4, 9 months n = 9, 12 months n = 9, >13 months n = 9 mice; cKO 2 months n = 8, 4 months n = 4, 9 months n = 8, 12 months n = 8, >13 months n = 5 mice). (Lower panels) Representative images of left ventricular M-mode echocardiography from WT compared to *Efnb1$^{CM-/-}$*KO mice, which show progressive LV dilatation and systolic function decline in the KO mice. (**B**) Kaplan–Meier survival plots for WT (blue) and *Efnb1$^{CM-/-}$*KO mice (red) (starting populations; WT n = 26, cKO n = 30 mice). Survival analysis was performed by log-rank test. (**C**) Heart weight/tibia length ratios from WT and *Efnb1$^{CM-/-}$*KO mice (WT 4 months n = 6, 9 months n = 8, 12 months n = 8 mice; cKO 4 months n = 6, 9 months n = 9, 12 months n = 8 mice). (**D**) CM area quantification from WGA-stained heart cross-sections (endocardium) from *Efnb1$^{CM-/-}$*KO or WT mice (~120 CMs/mouse; WT 2 months n = 5, 9 months n = 9, 12 months n = 8 mice; cKO 2 months n = 6, 9 months n = 9, 12 months n = 8 mice). (**E**) (Left panel) Cardiac fibrosis quantification from Masson's trichrome staining of transverse sections from WT and *Efnb1$^{CM-/-}$*KO mice hearts (2 months n = 6, 9 months n = 8, 12 months n = 9 mice) and (right panels) representative images. (**F**) Representative immunofluorescent staining of T-tubules (caveolin-3) in paraffin-embedded heart sections from *Efnb1$^{CM-/-}$*KO and WT mice. Data are presented as mean ± SD. One-way ANOVA with Tukey post-hoc test for longitudinal group comparisons of WT or cKO mice (each age compared with the preceding one), $^{\$}p<0.05$, $^{\$\$}p<0.01$, $^{\$\$\$}p<0.001$, $^{\$\$\$\$}p<0.0001$. Unpaired Student's *t*-test to compare WT and cKO groups. *p<0.05, **p<0.01, ***p<0.001, ****p<0.0001. Only significant results are presented.

The online version of this article includes the following source data for figure 8:

**Source data 1.** Raw data of echocardiographic measurements.

**Source data 2.** Raw data of Kaplan-Meier survival plots.

**Source data 3.** Raw data of heart weight/tibia length ratios.

**Source data 4.** Raw data of CM area quantification.

**Source data 5.** Raw data of cardiac fibrosis quantification.

*et al., 2001*; *Li et al., 1996*). Recent omics studies have detailed the molecular changes associated with the postnatal development of the mouse heart but only until P20 (*DeLaughter et al., 2016*; *Talman et al., 2018*). Thus, the P0-P20 postnatal development period essentially relies on the ECM remodeling of the cardiac tissue and, at the CM level, myofibril and Ca$^{2+}$ handling maturation, metabolic switch from glycolysis to major fatty acid oxidation and the transition from CM proliferation to CM hypertrophy (*Guo and Pu, 2020*; *Karbassi et al., 2020*). This metabolic change most likely occurs as an adaptation of the postnatal heart to cardiac tissue oxygenation and the newborn diet, which essentially consists of mother's milk and thus on high fatty acid availability. Our transcriptome analysis of mouse hearts reveals an additional developmental genetic program between P20 and P60 originating from all the cardiac cells of the left ventricle, thus indicating that the heart and the CM had not completed maturation by P20. Of interest, while the primary metabolic reliance of the heart on fatty acids and then glucose is already programmed between birth and P20, we found that the P20-P60 late postnatal stage programs the metabolic diversification of the heart. Such a new final metabolic adaptation of the heart coincides with the weaning process occurring around P20 in rodents, during which the newborn switches from milk to solid food (*Sengupta, 2013*). This step is also in line with the capacity of the heart to metabolize a large panel of substrates to meet the energy demands of the adult stage (*Murashige et al., 2020*). Remarkably, we found that the *Bdh1* gene was increased by more than 37-fold between P20 and P60, most likely indicating that the capacity of the heart to oxidize KB (*coming essentially from the liver*) as a fuel is not only restricted to the adaptation of the failing heart, as previously shown (*Horton et al., 2019*), but is also necessary for the adult physiological state. These results also agree with recent findings demonstrating that the human adult heart can use a large panel of substrates as a fuel, including a large proportion of KB (*Murashige et al., 2020*). Horton et al. did not report a major cardiac phenotype under basal conditions in CM-specific *Bdh1* KO mice (*Horton et al., 2019*). However, more in-depth cardiac phenotyping of these mice would undoubtedly be necessary to depict the role of KB in the adult cardiac physiology, such as diastolic function.

An intriguing finding from our study is the physiological cardiac hypertrophy of the heart between P20 and P60, which, from a CM lateral standpoint, does not rely on a classical myofibril addition in the CM but, at least in part, on an ephrin-B1-dependent- (1) crest maturation through SSM swelling at the CM surface and (2) lateral stretching of the CM. During postnatal development, heart growth occurs primarily through CM proliferation (hyperplasia) but transitions rapidly after birth (~P5-7) to CM growth (hypertrophy) (*Leu et al., 2001*; *Li et al., 1996*) through new myofibril biogenesis. Although physiological CM hypertrophy has been described beyond P20 and is related to increased cardiac mass (*Piquereau et al., 2010*; *Anversa et al., 1986*), the underlying molecular mechanisms have never been explored. *Piquereau et al., 2010* already reported and questioned such atypical

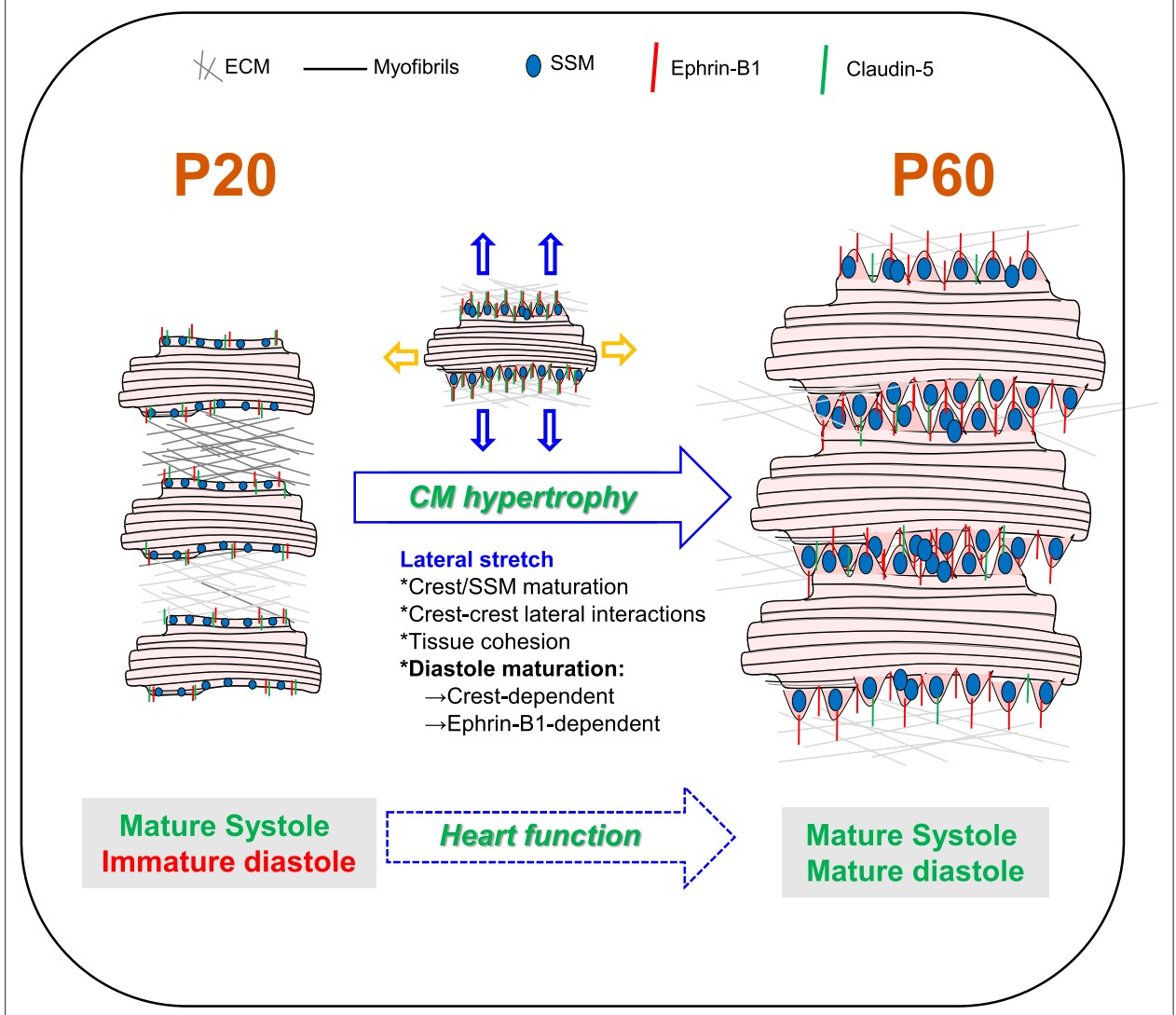

**Figure 9.** Schematic illustration of cardiomyocyte (CM) surface crest maturation between P20 and P60 supporting the setting of the adult diastolic function. At postnatal day 20 (P20), cardiomyocytes express low levels of ephrin-B1 and its claudin-5 partner at the lateral surface and surface crests/ subsarcolemmal mitochondria (SSM) are still immature. CMs do not interact with neighboring CMs at their lateral surfaces. After P20, CMs undergo substantial hypertrophy relying on elongation of both their long and short axes. From the short-axis standpoint, the hypertrophy is independent from myofibril addition but relies on the maturation of the CM surface architecture. Hence, at P60, the increase in ephrin-B1/claudin-5 expression at the lateral membrane promotes crest–SSM swelling but also crest–crest interactions between neighboring CMs most likely leading to a lateral stretch of the CM. This ephrin-B1-dependent lateral maturation of the CM surface allows maturation of the diastolic function.

hypertrophy of the rat heart after P21 without fiber addition, which was not classically related to the increase in heart weight to body weight index but conversely to a marked decrease. Here, we show that the P20-P60 hypertrophy relies on both an enlargement of the short and long axes of the CM. Interestingly, we demonstrated that the short-axis elongation is dependent on the architectural maturation of the CM surface trough an ephrin-B1 mechanism. Thus, the ephrin-B1 lateral membrane protein, a partner and regulator of claudin-5[9], participates in P20-P60 lateral CM hypertrophy, likely by bringing claudin-5 into the vicinity of neighboring CM, allowing intermittent but direct lateral crest–crest interactions (claudin-5/claudin-5 interactions), crest/SSM maturation, and the ensuing stretching of the CM lateral membrane (*Figure 9*). In agreement with this model, we previously showed that CMs from *Efnb1^{CM-/-}* KO mice exhibit substantially decreased levels of claudin-5 expression (*Genet et al., 2012*), likely accounting for the lack of crest maturation and lateral crest–crest interactions in these adult KO mice. However, other mechanisms also likely contribute to the short-axis elongation since it

was only partially prevented by ephrin-B1 deletion. More specifically, lateral membrane interactions with the ECM, i.e., through integrin or the dystroglycan complex, which we previously showed to be independent of ephrin-B1[9], could also play a role. In line with this assumption, our transcriptomic analysis identified the regulation of genes from the ECM remodeling pathway during the P20-P60 maturation period and the dystrobrevin-encoding gene (DTNA, *Figure 4—figure supplement 3*) from the dystroglycan complex. Apart from the short axis, P20-P60 CM hypertrophy also depends on the elongation of the CM long axis, which we show here to be independent from ephrin-B1 and which probably depends on the assembly of new sarcomeres at the myofibril extremities, thus contributing to the classical heart hypertrophy.

A new finding of our study is the specific maturation of SSM during the late P20-P60 postnatal period of the CM. Coinciding with this late cardiac mitochondria maturation in the CM, (*Piquereau et al., 2010*) previously reported a substantial increase in maximum respiratory capacity occurring after P21 and the adult stage in cardiac tissues from rats while the postnatal maturation of IFM occurred earlier. Biogenesis/maturation of cardiac mitochondria occur early during the embryogenesis, concurrently with the energy demand of the heart during the development period (*Lopaschuk and Jaswal, 2010*; *Zhao et al., 2019*.) Thus, they can adapt their morphology and function according to the energy need and metabolic conditions of the cell (*Packer, 1963*; *Schönfeld et al., 2000*). In this field, most of the works on the postnatal period have focused on the characterization of the IFM (the most abundant in the adult CM) or the global cardiac mitochondria activity dedicated to the energy supply to the CMs for the contractile machinery, without distinguishing the different mitochondria subpopulations of the adult CM (*Hollander et al., 2014*). IFM maturation during the P0-P7 stage occurs both through swelling and through architectural reorganization along the myofibrils (*Piquereau et al., 2010*), concomitantly with the well-known metabolic shift of the heart from glycolysis to a central oxidative metabolism more efficient for adult CM contraction (*Lopaschuk and Jaswal, 2010*). It follows that IFM function is primarily dedicated to supplying the CM with the energy necessary for adult contraction. The delayed maturation of SSM at the CM surface compared to intracellular IFM supports the concept that these mitochondria regulate different CM functions. In agreement with this notion, we demonstrated in this study that crests/SSM specifically regulate heart diastole. Several results support this conclusion: (1) young adult CM-specific *Efnb1* KO mice with immature SSM display diastolic defects with no impairment of systolic function, and (2) systolic function is already mature by P20 in rat hearts while the diastolic function is highly variable. Moreover, supporting our results about the late maturation of diastole, Zhou et al. previously reported that the diastolic function in the left ventricle of mice matures around the weaning period (*Zhou et al., 2003*). In the future, it would be interesting to investigate whether cardiac SSM defects are a specific feature of diastolic dysfunction pathologies, such as HFpEF frequently associated with a metabolic syndrome (*Valero-Muñoz et al., 2017*). In support of this concept, a recent study of myocardial gene expression signatures in human HFpEF reported that the ephrin-B1-encoding gene, which we demonstrated here as a specific determinant of the CM crests/SSM, is specifically downregulated in HFpEF heart patients compared with HFrEF (*Hahn et al., 2021*). Unfortunately, while several works have reported the influence of different cardiac pathologies on specific subpopulations of cardiac mitochondria (*Hollander et al., 2014*), these results require some caution, given the technical difficulty of specifically purifying and distinguishing the populations in the absence of reliable markers and due to the fact that these mitochondria are morphologically highly similar (*Guilbeau-Frugier et al., 2019*). Today, electron microscopy still remains the gold standard for accurately examining the cardiac mitochondria subpopulations.

An essential and new finding of our study is the identification of the ephrin-B1/crest/SSM module at the lateral membrane of the CM as a specific determinant of the physiological diastolic function of the adult heart. So far, the diastolic determinants have been proposed only in the context of diastolic dysfunction in cardiac pathologies such as HFpEF (*Lewis et al., 2017*). However, until rather recently, the lack of specific and compelling therapeutics in HFpEF underlines our limited knowledge of the control of diastole. This is overcomplicated in the context of HFpEF due to the contributions of several comorbidities delineating highly heterogeneous clinical features (*Lewis et al., 2017*). Here, our study demonstrates that *Efnb1* deletion specifically in the CM recapitulates at the young adult stage (2 months) some features of the diastolic dysfunction depicted in HFpEF, i.e., an elevated EDP and altered filling patterns combined with exercise intolerance. One important concern is understanding how ephrin-B1 at the lateral membrane of the adult CM can impact diastolic function. We previously

demonstrated that ephrin-B1 controls the architecture of the lateral membrane and the overall adult rod shape of the CM (*Genet et al., 2012*). Here, we now further demonstrated that ephrin-B1 controls the maturity of the crest/SSM architectural motif at the lateral membrane, thus playing a key role in the adult crest–crest interactions between neighboring CMs and in overall tissue cohesion. These lateral CM interactions likely dictate a mechanical lateral stretch of the CM, allowing perfect stacking between myofibril layers. Consequently, crest–crest interactions might contribute to controlling the relaxation length of the sarcomere, a mechanism that we previously suggested to be dependent on crest height (*Guilbeau-Frugier et al., 2019*). This ephrin-B1-dependent lateral stretch of the CM is also necessary for the spatial arrangement of the T-tubules and most likely its ensuing function during the P20-P60 development, as supported by the T-tubule disorganization in the CM-specific *Efnb1* KO mice. Thus, by altering the T-tubule structure, lack of ephrin-B1 could influence the $Ca^{2+}$ entry/$Ca^{2+}$ exit. It is worth noting that T-tubule disorganization in the CMs of $Efnb1^{CM-/-}$ KO mice is progressive with mild disorganization correlating with diastolic dysfunction only, while extensive disorganization is observed in HFrEF. Although defects in T-tubule architecture/function are a hallmark of HFrEF (*Guo et al., 2013*), it has been poorly examined in HFpEF, only one recent study reported that this mechanism is etiology-dependent (*Frisk et al., 2021*). An exciting feature of the CM-specific *Efnb1* KO mice is the progressive cardiac phenotype starting from a primarily diastolic defect that progressively switches toward a mild and then severe HFrEF and finally to death, thus highlighting the substantial cardioprotective role of the surface crests/SSM of the lateral membrane. Although this switch toward HFrEF is not a common feature of HFpEF patients (*Lupón et al., 2019*), definitive conclusions cannot be reached on the natural evolution of HFpEF, given that HFpEF patients received medication for their different comorbidities, which could protect them from HFrEF. In the future, an accurate characterization of the CM surface crests/SSM in different HFpEF models with diastolic dysfunction will undoubtedly shed light on whether crest disruption might be a hallmark of HFpEF and could thus contribute to the setting of the pathology.

## Materials and methods

### Animal models and euthanasia

CM-specific *efnb1* knock-out mice ($Efnb1^{CM-/-}$KO) have already been described (*Genet et al., 2012*). For generation of conditional CM-specific *efnb1* KO mice, $Efnb1^{flox/flox}$ mice (*Genet et al., 2012*) (obtained from Alice Davy) homozygous for the floxed *Efnb1* allele were bred to αMHC-Mer-Cre-Mer mice (RRID:IMSR_JAX:005657) directing expression of a tamoxifen-inducible Cre recombinase to CMs, to produce $Efnb1^{flox/flox}$-αMHC-Cre⁺ mice, i.e., *Efnb1* cKO mice. Deletion of exons 2–5 of the *Efnb1* gene was induced by intraperitoneal administration of 40 mg/kg/day tamoxifen (Sigma-Aldrich, Cat# T5648) for four consecutive days (*Figure 7—figure supplement 1*). $Efnb1^{flox/flox}$-αMHC-Cre⁻ mice similarly treated with tamoxifen were used as controls (Ctrl). WT, *efnb1* KO mice, Ctrl, and *efnb1* cKO mice were all kept in a mixed S129/S4×C57BL/6 background. All studies were performed on male and age-matched littermate mice. Studies on rats were performed on male Sprague–Dawley rats (purchased from Envigo, Huntingdon, UK; RGD_737903). Animals were euthanized with an intraperitoneal (i.p.) injection of 50 mg/kg *Dolethal* (Vetoquinol, Lure, France). All animal experiences were performed with the approval of the French CEEA-122 ethical committee (CEEA 122 2015–28).

### Transmission electron microscopy (TEM) and quantitative analysis

Heart samples were specifically processed to preserve CM surface crests as previously described (*Guilbeau-Frugier et al., 2019*). Thus, immediately after removal, the heart was washed in cold PBS for several seconds, then cautiously sliced (~1 mm³ biopsies) on a glass slide lying on an ice bed and fixed in 2% glutaraldehyde in Sorensen's buffer at 4°C and processed for TEM as previously described (*Dague et al., 2014*). Each tissue pieces was embedded in a different resin block. We first controlled for correct tissue orientation, i.e., with longitudinal fibers necessary to visualize the CM surface crests, through observation of semi-thin section stained with Toluidine blue with light microscopy. Then, two grids/resin block were proceeded. Next, we first evaluated the quality of the myocardium for each grid. TEM observation begins by checking for the absence of fixation artifacts based on mitochondria ultrastructure. When the fixation is not correct, the mitochondrial matrix appears clear and the cristae are disrupted and spaced out. In extreme cases, mitochondria are swollen and small clear spots can

be seen into the mitochondria or into the cytoplasm of CMs. These spots are larger than the lipid droplets and are randomly located within the cell. The observation of such fixation artifacts precludes the grid analysis. Moreover, sometimes CMs harbor contraction bands, reflecting a problem with cell relaxation at the time of animal sacrifice. Again, whether too many contraction bands are observed within a CM, it is not analyzed and we switch to the observation of another resin block. Generally, we analyzed ~10 CMs/grid if a specimen does not have 10 suitable CMs, another grid from another resin block is analyzed.

The observation goes on with an analysis of the orientation of CMs at low magnification (300×). A CM is correctly oriented for lateral crest observation if it is possible to see the staircase arrangement of the intercalated discs at the CM extremities and if the contractile apparatus of two adjacent longitudinal CMs is continuous. In the CM, sarcomeres should be aligned. It is preferable that the image section goes through the axis of the CM nucleus to reinforce the correct longitudinal orientation of the CM. The nucleus should be elongated with an indented electron-dense border. The IFM shape should appear elongated and mitochondria oriented along the contractile apparatus.

For quantifications, pictures are always taken at identical magnification. A CM is first photographed at low magnification (×300 or ×500, depending on the CM length) to attest to the quality and the orientation of the selected area. When crests can be visualized at the CM surface, then a crest is analyzed/quantified only if it is anchored on both sides of the Z-line. The crests must have a periodically repeated pattern. For each CM, several successive crests are photographed (each crest is photographed at ×7000 magnification) and subsequently quantified. Crest heights were quantified as described previously (*Guilbeau-Frugier et al., 2019*) relative to the sarcomere length to compensate for indifferent diastolic or systolic state of the cardiac tissue and taking into account that we have previously shown that crest height correlates with sarcomere length (*Guilbeau-Frugier et al., 2019*). The crest heights were measured manually from the top of the crest to the first myofibril top layer using Fiji software and the SSM number was quantified for each crest. The sarcomere lengths were measured similarly on the first myofibril layer below the CM surface crests. To analyze the interlateral space between two CMs, orientation of CMs in the longitudinal axis is mandatory. The selected area for quantification of the lateral space must not go through the crest architecture, so that the interlateral space appears regular and almost straight (in control conditions). The interlateral space is then photographed at high magnification (×5000) and five independent measurements were conducted. Myofibril numbers were quantified on the longitudinal axis of the CM at the level of their anchorage on an entire intercalated disk from top to bottom of the cell, thus allowing better individual counting of the myofibrils.

## Echocardiography, speckle tracking echocardiography/strain analysis, and Doppler imaging

All echocardiographies were performed under isoflurane anesthesia (1.5–2%). The animals' limbs were taped over the metal ECG leads. Body temperatures were carefully monitored and maintained as close to 37°C as possible during the entire procedure. For *Figure 2C* (P20 and P60 rats), transthoracic echocardiography was carried out using the Vevo 3100 (Fujifilm VisualSonics), equipped with a 21 MHz transducer (MX250). For *Figure 4A* (P20 and P60 rats), transthoracic echocardiography was carried out using the Acuson NX3 Elite (Siemens Healthineers) equipped with a high-frequency 16-MHz linear transducer (VF16-5 probe, Siemens Healthineers). For *Figure 6A* (adult mice), transthoracic echocardiography was carried out using a Vevo 2100 (Fujifilm VisualSonics), equipped with a 25 MHz transducer (MS250).

Left ventricular (LV) parasternal long-axis and short-axis 2D view in M-mode was performed at the papillary muscle to assess LV wall thicknesses and internal diameters. Left ventricular ejection fraction (LVEF) was also calculated (%) from a B-mode parasternal long-axis view by tracing endocardial end-diastolic and end-systolic borders to estimate LV volumes (EF % = [end-diastolic volume (EDV)-end-systolic volume]/EDV ×100).

The global longitudinal strain was measured from parasternal long-axis images using speckle-tracking-based imaging to evaluate global cardiac performance. Diastolic function was assessed by pulse-wave Doppler imaging of the mitral flow obtained from the apical four-chamber view. Early (E) and late (A) peak transmitral flow velocity and isovolumic relaxation time (IVRT) were measured, and the E/A ratio was calculated. Longitudinal motion of the mitral annulus was captured by tissue Doppler

imaging. Mitral annular velocities (E' and A' peaks) were measured and the E/A ratio and E/E' were calculated. Left atrium dimensions (anteroposterior diameter) were compared to the aortic diameter in the parasternal long axis. All measurements were quantified and averaged for three cardiac cycles and obtained by an examiner blinded to the genotype of the animals.

## Hemodynamic measurements of left ventricular pressure

Catheterization was performed before sacrifice under isoflurane anesthesia (2%). The animals were monitored with ECG leads and body temperatures were carefully monitored and maintained as close to 37°C as possible during the entire procedure. Hemodynamic parameters were measured using a Science pressure catheter (Transonic) connected to the PowerLab 8/35 data acquisition system (AD Instruments). The catheter was inserted into the right carotid artery, and after stabilization for 5 min, arterial blood pressure was recorded. After that, the catheter was advanced into the ascending aorta and then into the left ventricle under pressure control. After stabilization for 5 min, signals were continuously recorded. Trace analysis was then performed using the LabChart Pro software system (AD Instruments) to determine systolic and diastolic blood pressure, LV end-diastolic pressure (LVEDP), the maximum slope of the LV systolic pressure increment (dP/dtmax), and diastolic pressure decrement (dP/dtmin), and the relaxation time constant (Tau) was calculated from the LV pressure trace. All measurements were obtained by an examiner blinded to the genotype of the animals.

## Exercise exhaustion test

The maximum stress test with measurement of gas exchange was carried out according to the method described by *Kemi et al., 2002* using the TSE CaloTreadmill system, a fully computerized, electronically controlled system to investigate exercise calorimetry in mice and rats. CaloTreadmills are enclosed by an air-tight transparent housing that can be connected to a fully automated indirect gas calorimetry system operated by PhenoMaster software.

After 2 days of acclimatization to the treadmill, the stress test was carried out in mice from the two experimental groups, respecting heterogeneity between the two populations on each run. The mice warmed-up on an inclined treadmill (25°) at a speed of 5 cm.s$^{-1}$ for 5 min. Then, the mice ran at 15 cm.s$^{-1}$ for 5 min, and the speed was increased every 5 min by 5 cm.s$^{-1}$ until reaching 25 cm.s$^{-1}$. Finally, the speed was increased by 3 cm.s$^{-1}$ every 2 min until exhaustion. Exhaustion was defined as the inability of the mouse to return to running within 10 s of direct contact with an electric-stimulus grid. Calorimetric parameters, such as oxygen consumption (VO$_2$), carbon dioxide production (VCO$_2$), respiratory exchange rate (RQ), and energy expenditure (EE), are determined at fixed temporal intervals while the test animal is moving according to an experimenter-defined activity profile.

## Heart sections

For paraffin-embedded heart sections, hearts were excised and immediately fixed in 10% formalin (24 hr), 4% formalin (48 hr), and then embedded in paraffin and sectioned (transversal sections) at 5 µm intervals. For frozen cardiac tissue sections, hearts were embedded in OCT, frozen in liquid nitrogen, and 5 µm frozen sections were fixed with ice-cold acetone. as previously described (*Genet et al., 2012*).

## Immunohistochemistry and confocal imaging

Formalin-fixed paraffin-embedded or frozen cardiac tissue sections were used for immunohistochemical analysis. For paraffin-embedded sections, samples were deparaffinized, rehydrated, and subjected or not to heat-induced epitope retrieval depending on the antibodies. For cryosections, samples were permeabilized (0.5% Triton X100-PBS), blocked (Dako Protein Block, Dako, Cat# X0909 or 10% normal goat serum, Dako, Cat# X0907 or 0.5% BSA/PBS), and stained overnight at 4°C with the primary antibodies, then 1 hr RT with the fluorescent secondary antibodies all diluted in 0.1% BSA/0.1% Tween 20/0.5% Triton X-100-PBS. Nuclei were visualized with DAPI (Sigma-Aldrich, 32670). Cell membranes were stained with Oregon Green 488-conjugated or Texas Red-conjugated wheat germ agglutinin (WGA, Life Technologies, Cat# W6748 and W21405, respectively). All images were acquired on Zeiss LSM 780 confocal microscope using Zen 2011 software (Carl Zeiss).

## Antibodies for immunofluorescence studies

The primary antibodies in this study were as follows: goat anti-ephrin-B1 (R&D Systems; AF473); mouse anti-α sarcomeric actinin (Sigma-Aldrich, Cat# A7732, RRID:AB_2221571 or Abcam, Cat#

ab68167, RRID:AB_11157538); mouse anti-connexin 43 (Millipore; MAB3067); rabbit anti-N-cadherin (Epitomics; 2019-1); rabbit anti-desmoplakin 1/2 (ARP American Research Products, Cat# 03-61003, RRID:AB_1541118); rabbit anti-claudin-5 (Acris Antibodies, Cat# DP157, RRID:AB_978124); mouse anti-GAPDH (Abcam, Cat# ab9484, RRID:AB_307274), mouse anti-Ryanodine Receptor (Abcam, Cat# ab2868, RRID:AB_2183051), and mouse anti-caveolin-3 (BD Biosciences, Cat# 610420, RRID:AB_397800). The secondary fluorescent antibodies used in this study were as follows: donkey anti-goat Alexa Fluor488 (Molecular Probes, Cat# A-11055, RRID:AB_2534102); donkey anti-mouse Alexa Fluor488 (Cat# A-21202, RRID:AB_141607), goat anti-rabbit Oregon-Green488 (Cat# 011038), donkey anti-goat Alexa Fluor568 (Cat# A-11057, RRID:AB_2534104), and donkey anti-rat Alexa Fluor568 (Cat# A-11007, RRID:AB_10561522), all obtained from Thermo Fisher Scientific. Cell nuclei were stained with 4',6-Diamidino-2-phenylindole dihydrochloride-DAPI (Sigma-Aldrich, Cat# 32670) or TO-PRO–3 Iodide (642/661) (Invitrogen; Cat# T3605).

## 2D quantification of CM area/density

For in situ quantification of CM surface area and density, deparaffinized heart slides were stained with OG488-conjugated-WGA and CM area and density were measured in transversal heart cross-sections by manually tracing the cell contour on images of whole hearts acquired on a digital *NanoZoomer* (Hamamatsu) slide scanner using Zen 2011 software (Carl Zeiss). The experimenter was blinded to groups and genotypes.

## Quantification of cardiac fibrosis

Paraffin-embedded heart sections were stained using the Trichrome stain kit (Connective Tissue Stain, Abcam, ab150686). Fibrosis was quantified on Masson's trichrome-stained heart sections using an ImageJmacro from images of whole heart sections acquired on a digital *NanoZoomer* (Hamamatsu) slide scanner. The experimenter was blinded to the mouse genotype.

## Quantification of T-Tubules (TT) regularity in the cardiac tissue

TT architecture was quantified in LV myocardium from 4% PFA fixed-rat cardiac biopsies, and processed for caveolin-3 fluorescent immunostaining. Caveolin-3 was chosen as an indirect TT marker as it was previously shown that caveolin-3 and TT markers co-localized intracellularly at the Z-disk in adult CMs (*Ziman et al., 2010*). Images from caveolin-3-stained cardiac biopsies were acquired with a laser-scanning confocal microscope (Carl Zeiss LSM900), with ×63/1.4 oil objective and a 100 nm sampling. From these images, several regions of interest were extracted, each corresponding to different cells. This extraction was performed on ImageJ by manually rotating the image to have the CMs long axis oriented along the image horizontal axis. Then regions of interest with a fixed size of 500 × 35 pixels (corresponding to an area of 50 μm × 3.5 μm) were manually drawn and duplicated from the image. On these regions, the quantitative TT power measurement was performed with a Python script, written in a Jupyter Notebook. At first, a two-dimensional Fast Fourier Transformation (FFT2D) is performed. From the FFT2D, the power spectrum P_2D is calculated as: $P\_2D = |〚FFT〛\_2D|^2/N$ with N the total number of pixels.

From the power spectrum image, a one-dimensional power spectrum P_1D is retrieved by extracting the middle horizontal line. Finally, TT power (indicative of the regular organization of TT system) was measured on P_1D as the amplitude of a Gaussian curve fitted on a peak located between 0.45 μm⁻¹ and 0.7 μm⁻¹ and frequency indicative of as the center of the Gaussian curve. The script uses the libraries Numpy, Pandas, AICSImageIO, SciPy and Matplotlib.

## Cardiomyocyte isolation and contractility

Unloaded cell shortening and calcium transients were measured in freshly isolated ventricular myocytes, prepared as described previously (*Fauconnier et al., 2007*). Isolated cardiomyocytes were loaded with Indo-1 AM (10 μM, Invitrogen, Cat# I1223) at room temperature for 30 min and then washed out with the free HEPES-buffered solution (117 mM NaCl, 5.7 mM KCl, 4.4 mM NaHCO₃, 1.5 mM KH₂PO₄, 1.7 mM MgCl₂, 21 mM HEPES, 11 mM glucose, 20 mM taurine, pH 7.2) containing 1.8 mM Ca²⁺ (*Ait Mou et al., 2009*). Unloaded cell shortening and calcium concentration [Ca²⁺] (indo 1 dye) were studied using field stimulation (1 Hz, 22°C, 1.8 mM external Ca²⁺). Sarcomere length (SL) and fluorescence (405 and 480 nm) were simultaneously recorded (IonOptix System, Hilton, USA).

For other experiments, ventricular CMs were prepared from rats or mice as described before (*Genet et al., 2012*; *Dague et al., 2014*).

## Western blotting

For protein analysis from total cardiac extracts, cardiac tissue samples were directly lysed in RIPA buffer (25 mM Tris-HCl pH 7.6, 150 mM NaCl, 1% NP-40, 1% NaDOC, 0.1% SDS) in the presence of a protease inhibitor cocktail (Roche, Cat# 04693159001). The protein concentration of extracts was determined by the Lowry method (Bio-Rad) and equal amounts of proteins were subjected (50 µg) to SDS-PAGE and transferred to nitrocellulose membranes (Millipore). Proteins were detected with primary antibodies followed by horse radish peroxidase-conjugated secondary antibodies to goat (Thermo Fisher Scientific, Cat#31402, RRID: AB_228395) or rabbit (Sigma-Aldrich, Cat# NA934) IgG using an enhanced chemiluminescence detection reagent (GE Healthcare, Cat# RPN2235). Protein quantification was obtained by densitometric analysis using ImageQuant 5.2 software and was normalized to that of GAPDH expression and expressed in arbitrary units (A.U.).

## RNA extraction and microarray gene expression studies

RNA was extracted from left ventricle heart samples using the Tri-reagent method (*Chomczynski and Sacchi, 1987*) (Sigma-Aldrich, France). Gene expression profiles were performed at the GeT-TRiX facility (GénoToul, Génopole Toulouse Midi-Pyrénées) using Agilent Sureprint G3 Mouse GE v2 microarrays (8x60K, design 074809) following the manufacturer's instructions. For each sample, cyanine-3 (Cy3)-labeled cRNA was prepared from 200 ng of total RNA using the One-Color Quick Amp labeling kit (Agilent Technologies) according to the manufacturer's instructions, followed by Agencourt RNAClean XP (Agencourt Bioscience Corporation, Beverly, MA). Dye incorporation and cRNA yield were checked using the Dropsense 96 UV/VIS droplet reader (Trinean, Belgium). 600 ng of Cy3-labeled cRNA were hybridized on the microarray slides following the manufacturer's instructions. Immediately after washing, the slides were scanned on Agilent G2505C Microarray Scanner using the Agilent Scan Control A.8.5.1 software and fluorescence signal extracted using Agilent Feature Extraction software v10.10.1.1 with default parameters.

Microarray data and experimental details are available in NCBI's Gene Expression Omnibus (*Edgar et al., 2002*) and are accessible through GEO Series accession number GSE196257 (https://www.ncbi.nlm.nih.gov/geo/query/acc.cgi?acc=GSE196257).

## Microarray data statistical analysis

Microarray data were analyzed using R (R Core Team, 2018) and Bioconductor packages (*Huber et al., 2015*) as described in GEO accession GSE196257. Raw data (median signal intensity) were filtered, log2 transformed, and normalized using the quantile method (*Bolstad et al., 2003*).

A model was fitted using the limma lmFit function (*Ritchie et al., 2015*). Pair-wise comparisons between biological conditions were applied using specific contrasts. A correction for multiple testing was applied using the Benjamini–Hochberg procedure (*Klipper-Aurbach et al., 1995*) to control the false discovery rate (FDR). Probes with FDR ≤ 0.05 were considered to be differentially expressed between conditions.

Hierarchical clustering was applied to the samples and the differentially expressed probes using the Pearson correlation coefficient as distance and Ward's criterion for agglomeration. The clustering results are illustrated as a heatmap of expression signals. The enrichment of GO biological processes and KEGG pathways was evaluated using the enrichGO function and enrichKEGG functions from the clusterProfiler R package (*Yu et al., 2012*).

## Gene clustering according to the cardiac cell populations

Processed single-cell RNA-seq data from *Tucker et al., 2020* were retrieved from the Broad Institute's Single Cell Portal (study ID SCP498). Highly variable genes were selected on the scaled logNormalize dataset to perform cell clustering using Seurat 4.0.1 (using the default parameters in the FindVariableFeatures function in the Seurat R package). The uniform manifold approximation and projection (UMAP) were performed based on the first 30 principal components. Cell identities were made available by (*Tucker et al., 2020*) and used to color the cells on the UMAP plot.

## Data analysis: Statistics

The n number for each experiment and analysis is stated in each figure legend. The normality was tested using the Shapiro–Wilk normality test. Results are reported as mean ± SD. An unpaired Student's t-test for parametric variables was used to compare two groups. One-way ANOVA with Tukey post-hoc test was used for multiple group comparisons (>2 groups). Two-way ANOVA with Tukey post-hoc test was used for multiple group comparisons with two independent variables such as age (P20 and P60) and genotype (WT and cKO). The level of significance was assigned to statistics in accordance with their p values: *p≤0.05; **p≤0.01; ***p≤0.001; ****p≤0.0001). All graphs were generated using v9.2.0 (GraphPad Inc, San Diego, CA).

## Exclusion and inclusion criteria

Regarding transcriptomic data (*Figure 2B*), one WT P60 sample was excluded since it was, a posteriori, identified as a non-WT mouse sample. In *Figure 7D*, one Tau value was excluded from the cKO group since it was a negative value and thus an outlier.

## Acknowledgements

We thank Claire Naylies and Yannick Lippi (Toxalim, Université de Toulouse, INRAE, ENVT, INP-Purpan, UPS, Toulouse, France) for their contribution to microarray fingerprints acquisition and microarray data analysis carried out at GeT Genopole Toulouse Midi-Pyrénées facility (https://doi.org/10.15454/1.5572370921303193E12). We are also grateful to TRI Genotoul network facilities, specifically ANEXPLO platform (Toulouse) for help with echocardiography, the 'Centre de Microscopie Electronique Appliquée à la biologie-CMEAB' (Faculté de Médecine Rangueil-Toulouse) and the Cellular Imaging Facility-I2MC (Toulouse). This work was in part supported by the 'Fondation Bettencourt Schueller' (to CG), the 'Fondation de France' grant no. 75807 (to CG), the 'Fondation pour la Recherche Médicale' grant DEQ20170336733 (to CG) and grant FDM201906008682 (to BT) and the 'Société Francophone du Diabète' (to BT).

## Additional information

### Competing interests

Caroline Dubroca, Julile Maupoint, Thierry Sulpice: Dubroca, Julie Maupoint and Thierry Sulpice are employees of Cardiomedex. The other authors declare that no competing interests exist.

### Funding

| Funder | Grant reference number | Author |
|---|---|---|
| Fondation Bettencourt Schueller | - | Celine Galés |
| Fondation de France | 75807 | Celine Galés |
| Fondation Recherche Médicale | DEQ20170336733 | Celine Galés |

The funders had no role in study design, data collection and interpretation, or the decision to submit the work for publication.

### Author contributions

Clement Karsenty, Conceptualization, Data curation, Formal analysis, Validation, Investigation, Methodology, Writing – original draft, Writing – review and editing; Celine Guilbeau-Frugier, Conceptualization, Supervision, Validation, Investigation, Visualization, Methodology, Writing – original draft, Writing – review and editing; Gaël Genet, Conceptualization, Formal analysis, Investigation, Methodology, Writing – review and editing; Marie-Helene Seguelas, Romain Montoriol, Investigation; Philippe Alzieu, Supervision, Validation, Investigation; Olivier Cazorla, Conceptualization, Data curation, Formal analysis, Validation, Investigation, Writing – review and editing; Alexandra Montagner, Data curation, Formal analysis, Supervision, Validation, Investigation, Methodology,

Writing – review and editing; Yuna Blum, Conceptualization, Data curation, Validation, Investigation, Methodology; Caroline Dubroca, Formal analysis, Supervision, Validation, Investigation, Methodology, Writing – review and editing; Julile Maupoint, Formal analysis, Supervision, Validation, Investigation, Methodology; Blandine Tramunt, Investigation, Writing – review and editing; Marie Cauquil, Supervision, Investigation, Writing – review and editing; Thierry Sulpice, Supervision, Funding acquisition, Writing – review and editing; Sylvain Richard, Data curation, Formal analysis, Investigation, Writing – review and editing; Silvia Arcucci, Data curation, Formal analysis, Validation, Investigation, Writing – review and editing; Remy Flores-Flores, Validation, Investigation, Methodology; Nicolas Pataluch, Data curation, Investigation; Pierre Sicard, Data curation, Validation, Investigation; Antoine Deney, Formal analysis, Validation, Investigation; Thierry Couffinhal, Data curation, Formal analysis, Supervision, Validation, Writing – review and editing; Jean-Michel Senard, Conceptualization, Formal analysis, Supervision, Validation, Methodology, Writing – review and editing; Celine Galés, Conceptualization, Data curation, Formal analysis, Supervision, Funding acquisition, Validation, Investigation, Methodology, Writing – original draft, Project administration, Writing – review and editing

### Author ORCIDs

Gaël Genet http://orcid.org/0000-0003-0016-5887
Pierre Sicard http://orcid.org/0000-0001-5837-3916
Jean-Michel Senard http://orcid.org/0000-0001-7679-1281
Celine Galés http://orcid.org/0000-0002-4938-1583

### Ethics

Experimental animal protocols were carried out in accordance with the French regulation guidelines for animal experimentation and were approved by the French CEEA-122 ethical committee. Animals were euthanized with an intraperitoneal (i.p.) injection of 50 mg/kg Dolethal (Vetoquinol, Lure, France).

### Decision letter and Author response

Decision letter https://doi.org/10.7554/eLife.80904.sa1
Author response https://doi.org/10.7554/eLife.80904.sa2

## Additional files

### Supplementary files

• MDAR checklist

### Data availability

Transcriptomic data have been deposited in GEO under the accession number GSE196257. Source data files have been provided for Figures 1B, C, 3 A, B, 4 A-G, 5 A-G, 6 A-D, 7B-D, 8 A-E and Figure 1-figure supplement 5, 8C, Figure 4-figure supplement 1, 2, 3 A-B, Figure 7-figure supplement 1.

The following dataset was generated:

| Author(s) | Year | Dataset title | Dataset URL | Database and Identifier |
|---|---|---|---|---|
| Clément K, Yannick L, Naylies C, Montagner A, Galés C | 2022 | Global cardiac gene expression in left ventricles from 20 day-old and 60 day-old Efnb1 CMspe KO and WT male mice | https://www.ncbi.nlm.nih.gov/geo/query/acc.cgi?acc=GSE196257 | NCBI Gene Expression Omnibus, GSE196257 |

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
