## [Editor Report]

This article will be of interest to the field of cardiac development and function. It provides detailed characterization of the cardiomyocyte surface crests, and also provides evidence that loss of Ephrin-B1 leads to compromised surface crest formation and diastolic dysfunction. The article did not provide direct evidence that the diastolic dysfunction of Ephrin-B1 cardiac mutant is primarily attributable to the compromised postnatal maturation of cardiomyocyte surface crests; however, this work and the rigor applied represent an important hypothesis and contribution to the field.

---

## [Decision Letter]

**Decision letter after peer review:**

Thank you for submitting your article "Ephrin-B1 regulates the adult diastolic function through a late postnatal maturation of cardiomyocyte surface crests" for consideration by *eLife*. Your article has been reviewed by 5 peer reviewers, one of whom is a member of our Board of Reviewing Editors, and the evaluation has been overseen by a Reviewing Editor and a Senior Editor. The following individual involved in review of your submission has agreed to reveal their identity: Stephane Hatem (Reviewer #3).

Essential revisions:

1) Please address the comments raised by the reviewers, particularly reviewers #2, #3, and 4#.

*Reviewer #2 (Recommendations for the authors):*

1. Authors should clearly state the genotypes of the mice used, and whether controls are littermates or not. The word littermate does not appear in the paper, so I am guessing they are not, in which case the mouse experiments must be repeated with appropriate controls.

2. The authors appear to conflate a developmental phenotype that ephrin-b1 appears to be involved with vs HFpEF, which is typically an acquired human disease. An inducible, conditional-knockout (Mer-Cre-Mer) for example would be the appropriate model to determine that ephrin-b1 is linked to HFpEF. Frankly, their prior manuscript in Cardiovascular Res could have linked this pathway to more of a HFpEF phentype using their pressure overload/TAC model. Specifically, the authors didn't observe a major decline in ejection fraction in those animals and it seems like the diastolic parameters simply were not examined. If the echo data are available, a re-examination of those data could provide more evidence of diastolic dysfunction. In any case, the ephrin-b1 mice examined in this paper go on to develop significant systolic dysfunction and mortality, which is unlikely to be due to the mild diastolic abnormalities presented in the manuscript.

*Reviewer #3 (Recommendations for the authors):*

1. Concerning the maturation of crest at P20 and P60 which might be related, at least partly, to changes in myocardial working conditions, I wonder if this ultrastructural organization changes between cardiac areas such as septum, apex, out-put track, left and right ventricle?

2. Is the maturation of the diastolic function between P20 and P60 associated with changes in the expression in gene encoding for proteins of the EC coupling notably SERCA and phsopholamban (volcano plot)?

3. Could you comment, igure 3 B, is the increase in blood pressure caused by an increased cardiac output while peripheral resistance are unchanged?

4. A major finding of the study is that the protein ephrin-B1 plays a key role in adult crest-crest interactions between CM and that appears to be a major determinant of the normal diastolic cardiac function. Therefore, it is tempting to conclude that this process contributes to HFpEF. However, in my opinion this conclusion could be strengthened by providing evidence, if available, for alteration of crest-crest interaction and/or ephrin B1 expression in well-established model of HFpEF.

5. Figure 7 E, cross section of heart barely illustrated the accumulation of fibrosis and seems to indicate some atrophy of ventricle compared to Wt in contraction with the rest of the study.

6. Given the presence in this crest region of subpopulations of ionic channels notably sodium ones (Eichel et al. Circ Res 2016), I wonder if there any modification of the ECG of KO mice?

7. The section of the discussion on empaglifozine (page 17) is not supported by results and appears as out of the scope of the study.

*Reviewer #4 (Recommendations for the authors):*

As the main proof of the existence of p60 stage comes from the morphometric analysis of TEM images I recommend making an effort of explaining in minute details how the analysis has been done. How the randomization of images and fields of view has been achieved? How many cells were analysed, how many blocks containing tissues from different animals? How many random fields of view? What statistical methods were used to validate differences?

*Reviewer #5 (Recommendations for the authors):*

Cardiac maturation, which occurs in the neonatal stage, is the last phase of heart development that prepare the heart for stronger and more efficient pumping through postnatal life. This cardiac maturation process is characterized by many molecular, structural, and functional changes as the heart transitions from a fetal to an adult phenotype. In this study, Karsenty et al., characterized a late postnatal cardiac maturation event- the formation of a lateral membrane surface architecture with periodic crests – that occurs around p20-p60, a time window that the authors claim to be that for maturation of the diastolic function. Further study suggests that CM specific Efnb1 knockdown impairs the late maturation of surface crests and the diastolic function.

Specific comments:

1. The images in Figure 1A and supplementary Figure 1 don't show any obvious sarcomeres and mitochondria. Although cardiac maturation features myofibril expansion during neonatal stages, sarcomere formation is initiated during fetal developmental stages and should be visible at P1 (see examples in Gong et al., PMID: 26785495).

2. the transcriptomic study and GO analysis don't provide any direct evidence supporting p20-p60 stage dedicated to the development of the surface crests and diastolic function. The upregulated GO terms are mostly associated with the process of immune defense, angiogenesis, positive regulation of cell death, which are irrelevant to surface crests and diastolic function. Even the GO term muscle cell differentiation is too general to provide any specific information. Diastolic stiffness of the LV is in large part dependent on the N2BA:N2B isoform ratio. How does the N2BA:N2B ratio change during postnatal cardiac maturation, and does the change correlate with the maturation of diastolic function?

3. The authors claim in the manuscript that "this maturation step of the crests is ephrin-B1-dependent and specifically regulates the diastolic function of the adult heart". Yet, although crest maturation of maturation of diastolic function coincides temporally, there is direct evidence that maturation step of the crests regulates cardiac diastolic function.

---

## [Author Response]

Reviewer #2 (Recommendations for the authors):1. Authors should clearly state the genotypes of the mice used, and whether controls are littermates or not. The word littermate does not appear in the paper, so I am guessing they are not, in which case the mouse experiments must be repeated with appropriate controls.

All control, wild-type (WT) mice used in this paper were always littermates of Efnb1 KO mice. We agree that this is a major detail that has been forgotten in the initial version of our manuscript but it has been now included in the “materiel and methods/Animal models and euthanasia“ section.

2. The authors appear to conflate a developmental phenotype that ephrin-b1 appears to be involved with vs HFpEF, which is typically an acquired human disease. An inducible, conditional-knockout (Mer-Cre-Mer) for example would be the appropriate model to determine that ephrin-b1 is linked to HFpEF. Frankly, their prior manuscript in Cardiovascular Res could have linked this pathway to more of a HFpEF phentype using their pressure overload/TAC model. Specifically, the authors didn't observe a major decline in ejection fraction in those animals and it seems like the diastolic parameters simply were not examined. If the echo data are available, a re-examination of those data could provide more evidence of diastolic dysfunction. In any case, the ephrin-b1 mice examined in this paper go on to develop significant systolic dysfunction and mortality, which is unlikely to be due to the mild diastolic abnormalities presented in the manuscript.

First at all, we have to mention that the first instance purpose of our manuscript was not at all to claim that ephrin-B1 is linked to HFpEF but that CM crests/SSM and their molecular determinants*, i.e.* claudin-5 and ephrin-B1, are linked to the physiological postnatal maturation of the CM and the setting of the diastolic function.

Now, we agree that we showed that the constitutive *Efnb1* CM specific-KO model lacking ephrin-B1 in the CM reminisces, at the adult stage, some of the HFpEF features in patients, but which doesn’t mean at all that this mouse model is an HFpEF model. This aspect will need much more specific exploration and will be the next step following this current paper with the aim to characterize the potential loss of CM surface crests in different validated HFpEF models, as suggested by the reviewer with the TAC model (we didn’t explore the diastolic function at all in this study). This will be the take home message of a next story focusing of the pathophysiology of HFpEF (currently under process).

Now, we completely agree with the reviewer that we cannot completely exclude that the phenotype of the constitutive Efnb1 KO mice we described in the current paper is related to other postnatal developmental stage instead of pure CM surface crest disruption at the adult stage. Also, to discriminate between these two possibilities, we have now used in the revision process a tamoxifen-inducible conditional-knockout (Mer-Cre-Mer) of *Efnb1* in the CM (aMHC promotor). This mouse model has never been reported before but its characterization (new Figure 7—figure supplement 1) indicated that tamoxifen injection can lead to more than 50 % of Efnb1 deletion specifically in CMs. In these conditions, deletion of *Efnb1* (tamoxifen injection) was initiated at the young adult stage (2 month-old) and the systolic and diastolic function (echo Doppler and LV-catheterism) but also CM crest phenotype (TEM) were examined 1 month later. As shown in the new Figure 7, deletion of *efnb1* at the adult stage led to partial loss of CM surface crests (New Figure 7A), agreeing with the partial deletion of *Efnb1*, associated with a significant increase in the IVRT (echo-doppler), LVEDP (LV catheterism) with no modification of the ejection fraction (echo) compared to the control mouse littermate (tamoxifen injected) (New Figure 7B, C). Thus, these data clearly demonstrate that Ephrine-B1 is a specific determinant of the crest architecture at the CM surface and of the diastolic function at the adult stage.

Reviewer #3 (Recommendations for the authors):1. Concerning the maturation of crest at P20 and P60 which might be related, at least partly, to changes in myocardial working conditions, I wonder if this ultrastructural organization changes between cardiac areas such as septum, apex, out-put track, left and right ventricle?

The reviewer is completely right. Last year, we have started exploring the maturation of the different heart compartments between P20 and P60 (left/right atria, left/right ventricles, infundibulum, septum, apex, cardiac area close to the valves). This work is still under process and will be the topic of a new article, but the first results indicated major differences in unexpected compartments like the apex.

2. Is the maturation of the diastolic function between P20 and P60 associated with changes in the expression in gene encoding for proteins of the EC coupling notably SERCA and phsopholamban (volcano plot)?

We didn’t notice any modifications of genes of the EC coupling in our transcriptomic data between P20 and P60. Which doesn’t mean that the related proteins are well localized within the CM, as it is for claudin-5 or Ephrin-B1, but we didn’t explore it. We have just explored the maturation of the T-Tubules in this paper.

3. Could you comment, igure 3 B, is the increase in blood pressure caused by an increased cardiac output while peripheral resistance are unchanged?

We think that it is due to both. Cardiac afterload is increasing since arterial pressure increased between P20 and P60 probably due to an increase in peripheral resistance (known in human but not proven in murine model). Cardiac output is also increasing during physiological development (*doi: 10.1152/ajpheart.2000.278.2.H652*.) but finally cardiac index remains stable. To go further, we don’t know if inotropy is increasing, but cardiac PV loop experiments could help answering this question, although technically challenging in murine models.

4. A major finding of the study is that the protein ephrin-B1 plays a key role in adult crest-crest interactions between CM and that appears to be a major determinant of the normal diastolic cardiac function. Therefore, it is tempting to conclude that this process contributes to HFpEF. However, in my opinion this conclusion could be strengthened by providing evidence, if available, for alteration of crest-crest interaction and/or ephrin B1 expression in well-established model of HFpEF.

The reviewer is completely right. But as already explain to the reviewer 2 (answer 2), the first instance purpose of our current manuscript was not at all to claim that ephrin-B1 is linked to HFpEF but that CM crests/SSM and their molecular determinants*, i.e.* claudin-5 and ephrin-B1, are linked to the physiological postnatal maturation of the CM and the setting of the diastolic function.

Now, we agree that we showed that the constitutive *Efnb1* CM specific-KO model lacking ephrin-B1 in the CM reminisces, at the adult stage, some of the HFpEF features in patients, but which doesn’t mean at all that this mouse model is an HFpEF model. This aspect will need much more specific exploration and will be the next step following this current paper with the aim to characterize the potential loss of CM surface crests in different validated HFpEF models, as suggested by the reviewer with the TAC model (we didn’t explore the diastolic function at all in this study). This will be the take-home message of a next story focusing of the pathophysiology of HFpEF (currently under process). We do think that this manuscript already includes a lot of data (late postnatal maturation of the heart and the role of the crest maturation in the diastolic function) and are afraid that introducing the pathophysiological aspect will further increase the complexity and dilute the take-home message.

However, in the current version of our manuscript, we cannot completely exclude that the phenotype of the constitutive Efnb1 CM-KO mice we described at the adult stage is directly related to specific alteration of CM surface crest/diastolic function at the adult stage or more likely related to other earlier developmental defects (secondary mechanisms). Also, to discriminate between these two possibilities, we have now used a tamoxifen-inducible conditional-knockout (Mer-Cre-Mer) of *Efnb1* in the CM (aMHC promotor). This mouse model has never been reported before but its characterization (new Figure 7—figure supplement 1) indicated that tamoxifen injection can lead to up to 50 % of Efnb1 deletion in CMs. In these conditions, deletion of *Efnb1* (tamoxifen injection) was initiated at the young adult stage (2-month old) and the systolic and diastolic function (echo Doppler and LV-catheterism) but also CM crest phenotype (TEM) were examined one month later. As shown in the new Figure 7, deletion of *efnb1* at the adult stage led to partial loss of CM surface crests (New Figure 7B), agreeing with the partial deletion of *Efnb1*, associated with a significant increase in the IVRT (echo-doppler), LVEDP (LV catheterism) with no modification of the ejection fraction (echo) compared to the control mouse littermate (tamoxifen injected) (New Figure 7C, D). Thus, these data clearly demonstrate that ephrin-B1 is a specific determinant of the crest architecture at the CM surface and of the diastolic function at the adult stage. These new results showing the direct link between Crests/ephrin-b1 maturity and the diastolic function clearly support the following step, i.e. to explore in the future the crest phenotype in the pathophysiology of HFpEF.

5. Figure 7 E, cross section of heart barely illustrated the accumulation of fibrosis and seems to indicate some atrophy of ventricle compared to Wt in contraction with the rest of the study.

Fibrosis was only visible in the cKO 12 months (blue staining). There is no atrophy of the ventricles between KO and WT mice but for better comparison, we have also repositioned all WT and KO heart’s images in the same orientation.

6. Given the presence in this crest region of subpopulations of ionic channels notably sodium ones (Eichel et al. Circ Res 2016), I wonder if there any modification of the ECG of KO mice?

The reviewer is completely right. We have currently a rhythmology paper under process that has fully explored the ECG in adult *Efnb1* KO and WT mice (basal and triggered arrhythmia). This study indicates that only the PR interval was significantly decreased in 2-month old KO versus WT mice (KO=35,9 ±3,3 vs WT=39,3± 3,3; n=48 mice/group). This shorter PR interval reminisces the physiological PR interval of children (shorter PR interval than adults) and thus completely agree with our current paper showing that the adult *Efnb1*-CM spe KO mice harbor some pediatric features due to late developmental defects, i.e. the lack of P20-P60 maturation of the lateral membrane.

7. The section of the discussion on empaglifozine (page 17) is not supported by results and appears as out of the scope of the study.

We agree with the reviewer. We thus removed this discussion.

Reviewer #4 (Recommendations for the authors):As the main proof of the existence of p60 stage comes from the morphometric analysis of TEM images I recommend making an effort of explaining in minute details how the analysis has been done. How the randomization of images and fields of view has been achieved? How many cells were analysed, how many blocks containing tissues from different animals? How many random fields of view? What statistical methods were used to validate differences?

We agree with the author that TEM is often associated with “all sorts of oddities” and that‘s the reason our recent paper (*Guilbeau-Frugier et al., Cardiovasc Research* 2019) was dedicated to the analysis of technical pitfalls and analysis. All this paper relies on that: How to proceed the cardiac tissue to avoid artifacts on the crests/SSM visualization and how to quantify them?.

Now, instead of only citing our previous paper, we have implemented the *“Material and methods” / “Transmission electron microscopy (TEM) and quantitative analysis”* section (Main manuscript, page 20-21) by highly detailing all the TEM observation/quantification.

The question of randomization of images of the number fields of view is a general question in all imaging techniques and not specific at all with our TEM study. In imaging, there is no randomization.

All statistical analysis of TEM data quantifications are accurately described in all figure legends. For instance, in the figure 1: (B) Quantification of crest heights / sarcomere length (left panel), SSM number / crest (middle panel) and SSM area (right panel) from TEM micrographs obtained from P20- or P60 rat hearts (P20 n=6, P60 n=6; 4 to 8 CMs/rat, ~ 70 crests/rat). However, to better clarify the “P20 n=6, P60 n=6”, we have now specified “P20 or P60 n=6 rats”. This have been now specified in the figure legends for all statistical analysis (highlighted in yellow in the revised manuscript).

Reviewer #5 (Recommendations for the authors):Cardiac maturation, which occurs in the neonatal stage, is the last phase of heart development that prepare the heart for stronger and more efficient pumping through postnatal life. This cardiac maturation process is characterized by many molecular, structural, and functional changes as the heart transitions from a fetal to an adult phenotype. In this study, Karsenty et al., characterized a late postnatal cardiac maturation event- the formation of a lateral membrane surface architecture with periodic crests – that occurs around p20-p60, a time window that the authors claim to be that for maturation of the diastolic function. Further study suggests that CM specific Efnb1 knockdown impairs the late maturation of surface crests and the diastolic function.Specific comments:1. The images in Figure 1A and supplementary Figure 1 don't show any obvious sarcomeres and mitochondria. Although cardiac maturation features myofibril expansion during neonatal stages, sarcomere formation is initiated during fetal developmental stages and should be visible at P1 (see examples in Gong et al., PMID: 26785495).

Both Figure 1A and Figure 1—figure supplement 1 show both sarcomere (especially the Z lines since sarcomere are not at all completely established) and mitochondria. However, because the initial images were a bit too dark, we have changed the contrast/luminosity for better appreciation of the cellular details.

2. The transcriptomic study and GO analysis don't provide any direct evidence supporting p20-p60 stage dedicated to the development of the surface crests and diastolic function. The upregulated GO terms are mostly associated with the process of immune defense, angiogenesis, positive regulation of cell death, which are irrelevant to surface crests and diastolic function. Even the GO term muscle cell differentiation is too general to provide any specific information. Diastolic stiffness of the LV is in large part dependent on the N2BA:N2B isoform ratio. How does the N2BA:N2B ratio change during postnatal cardiac maturation, and does the change correlate with the maturation of diastolic function?

We totally agree with the reviewer regarding transcriptomic study and GO analysis don't provide any direct evidence supporting p20-p60 stage dedicated to the development of the surface crests and diastolic function. This experimental part was not at all on this purpose but to provide evidence for a global transcriptomic change of the whole heart between P20 and P60 since most of the papers in the postnatal field assume that the postnatal developmental stage of the heart is completed at P20 due to the presence of the mature rod-shape of the CM. These results thus indicate that a new development stage occurs between P20 and P60.

Now, we don’t agree with the reviewer with its statement that surface crest or diastolic function cannot be related to GO pathways like immune defense, angiogenesis, positive regulation of cell death. Nobody knows at the moment the molecular determinants involved on the control of the diastole. In diastolic defects associated with HFpEF this is even more complex given the pluridimensional origin of this pathology. Defects in the microcirculation have been described in HFpEF which could be related to the angiogenesis pathways for instance. We can just say that it is impossible from the GO analysis to extract an intuitive pathway associated with the crest/diastole but again this was not the objective of this analysis. In this context of multiple specific defects reported in diastolic dysfunction models, we thus do think that examining the N2BA:N2B isoform ratio will not bring a better link between crest and the diastolic function.

3. The authors claim in the manuscript that "this maturation step of the crests is ephrin-B1-dependent and specifically regulates the diastolic function of the adult heart". Yet, although crest maturation of maturation of diastolic function coincides temporally, there is direct evidence that maturation step of the crests regulates cardiac diastolic function.

In our manuscript, we started with comparison of the cardiac tissue between P20 an P60 and showing several concomitant events including ephrin-B1/Claudin-5 trafficking at the CM lateral surface, CM crest maturation and maturation of the diastolic function. Thus, at this experimental stage, these events are just correlated and obviously not a criterion of any relationship between these two events.

To better gain insight into a role for the CM surface crests in the maturity of the diastolic function, we have next deleted another potential molecular determinant of the crests (claudin-5 partner), *i.e*, Ephrin-B1, through the use of the embryonic *Efnb1-*CM specific-KO model. We showed that lack of ephrin-B1 in the CM led to a lack in crest maturation and diastolic defect, thus indicating that at least ephrin-B1 is specifically involved, directly or indirectly we don’t know, in the maturation process of the crests in the CM.

Now, we completely agree with the reviewer that we cannot completely exclude that the crest/diastole phenotype of the constitutive *Efnb1*-CM spe KO mice we have described at the adult stage in the initial version of the paper is indeed the consequence of another earlier developmental defect and not specific to the CM crest defects at the adult stage.

Also, to discriminate between these two possibilities, we have now used in the revision process a tamoxifen-inducible conditional-knockout (Mer-Cre-Mer) of *Efnb1* specifically in the CM (aMHC promotor). This mouse model has never been reported before but its characterization (new Figure 7—figure supplement 1) indicated that tamoxifen injection can lead to up to 50 % of Efnb1 deletion in CMs. In these conditions, deletion of *Efnb1* (tamoxifen injection) was initiated at the young adult stage (2-month old) and the systolic and diastolic function (echo Doppler and LV-catheterism) but also CM crest phenotype (TEM) were examined one month later. As shown in the new Figure 7, deletion of *efnb1* at the adult stage led to partial loss of CM surface crests (New Figure 7B), agreeing with the partial deletion of *Efnb1*, associated with a significant increase in the IVRT (echo-doppler), LVEDP (LV catheterism) with no modification of the ejection fraction (echo) compared to the control mouse littermate (tamoxifen injected) (New Figure 7C, D). Thus, these data clearly demonstrate that ephrin-B1 is a specific determinant of the crest architecture at the CM surface and of the diastolic function at the adult stage.